# Source specific bias correction of US background and anthropogenic ozone modeled in CMAQ

T. Nash Skipper[1,*], Christian Hogrefe[2], Barron H. Henderson[2], Rohit Mathur[2], Kristen M. Foley[2], Armistead G. Russell[1]

[1]School of Civil & Environmental Engineering, Georgia Institute of Technology, Atlanta, GA 30332, USA
[2]U.S. Environmental Protection Agency, Research Triangle Park, NC, 27709, USA
[*]Now at U.S. Environmental Protection Agency, Research Triangle Park, NC, 27709, USA

*Correspondence to*: Armistead G. Russell (ar70@gatech.edu)

**Abstract.** United States (US) background ozone ($O_3$) is the counterfactual $O_3$ that would exist with zero US anthropogenic emissions. Estimates of US background $O_3$ typically come from chemical transport models (CTMs), but different models vary in their estimates of both background and total $O_3$. Here, a measurement-model data fusion approach is used to estimate CTM biases in US anthropogenic $O_3$ and multiple US background $O_3$ sources, including natural emissions, long-range international emissions, short-range international emissions from Canada and Mexico, and stratospheric $O_3$. Spatially and temporally varying bias correction factors adjust each simulated $O_3$ component so that the sum of the adjusted components evaluates better against observations compared to unadjusted estimates. The estimated correction factors suggest a seasonally consistent positive bias in US anthropogenic $O_3$ in the eastern US, with the bias becoming higher with coarser model resolution and with higher simulated total $O_3$ though the bias does not increase much with higher observed $O_3$. Summer average US anthropogenic $O_3$ in the eastern US was estimated to be biased high by 2, 7, and 11 ppb (11%, 32%, and 49%) for one set of simulations at 12, 36, and 108 km resolutions and 1 and 6 ppb (10% and 37%) for another set of simulations at 12 and 108 km resolutions. Correlation among different US background $O_3$ components can increase the uncertainty in the estimation of the source-specific adjustment factors. Despite this, results indicate a negative bias in modeled estimates of the impact of stratospheric $O_3$ at the surface, with a western US spring average bias of -3.5 ppb (-25%) estimated based on a stratospheric $O_3$ tracer. This type of data fusion approach can be extended to include data from multiple models to leverage the strengths of different data sources while reducing uncertainty in the US background ozone estimates.

# 1 Introduction

United States (US) background ozone ($O_3$) is the counterfactual $O_3$ that would exist if US
anthropogenic emissions were zero. The National Ambient Air Quality Standard (NAAQS) for $O_3$ was
set at a level of 70 ppb in 2015 and may be lowered. In its recent reviews of the $O_3$ NAAQS, the US
Environmental Protection Agency (EPA) noted the importance of US background $O_3$ (US EPA, 2013,
2014, 2020b, a). US background $O_3$ takes up a larger portion of the allowed ozone as the NAAQS is
tightened and is a larger portion of total observed $O_3$ as anthropogenic precursor emissions decline (Lin
et al., 2017; Guo et al., 2018; Jaffe et al., 2018). US background $O_3$ cannot be observed (Fiore et al., 2003;
Dentener et al., 2010; Mcdonald-Buller et al., 2011; Fiore et al., 2014; Jaffe et al., 2018; US EPA, 2013,
2014, 2020b, a). It is typically quantified using a chemical transport model (CTM), most commonly using
the zero-out method in which US anthropogenic emissions are set to zero. There is much uncertainty in
CTM estimates of US background $O_3$ due to model biases and differences in CTM-estimated US
background $O_3$ among different models (Mcdonald-Buller et al., 2011; Fiore et al., 2014; Dolwick et al.,
2015; Huang et al., 2015; Guo et al., 2018; Jaffe et al., 2018). Jaffe et al. (2018) estimated that the typical
uncertainty in CTM-estimated seasonal mean US background $O_3$ is $\pm 10$ ppb.

Sources of US background $O_3$ include naturally occurring emissions such as wildfires, biogenic
VOCs, oxides of nitrogen ($NO_x$) from soil, lightning $NO_x$, stratosphere-to-troposphere exchange, and
oxidation of methane (Fiore et al., 2014; Jaffe et al., 2018; US EPA, 2020a). Some portions of total $O_3$
contributions from soil $NO_x$ and methane oxidation are US background sources while some are
anthropogenic. Soil $NO_x$ is emitted by microbial processes in both natural and agricultural lands and is
limited by availability of nitrogen in the soil. There is a pre-industrial level of methane that contributes to
US background $O_3$ formation, but any $O_3$ created through oxidation of methane above the pre-industrial
level is anthropogenic. Soil $NO_x$ and methane oxidation are often treated as US background $O_3$ sources
in their entirety in CTM studies due to the complexity of splitting up the natural and anthropogenic
portions (US EPA, 2020a). Wildfires are treated as US background $O_3$ sources, but the impacts of
wildfires on $O_3$ can be affected by US anthropogenic emissions when VOCs from fires are transported
over $NO_x$-rich urban areas, leading to enhanced $O_3$ production (Jaffe et al., 2013; Langford et al., 2023;
Rickly et al., 2023). US background $O_3$ sources also include non-US anthropogenic pollution which may

be from long range transport (Lin et al., 2012b) or from short range transport from neighboring countries (Wang et al., 2009).

In previous work (Skipper et al., 2021), we developed a bias correction method which used regression modeling to adjust CTM-simulated US anthropogenic and US background $O_3$ to better align with observations and to improve agreement of differing US background $O_3$ estimates from different model configurations. We developed spatially and temporally varying scaling factors to adjust US anthropogenic and US background $O_3$. In that work, US background $O_3$ was treated as a single quantity rather than considering different sources of US background $O_3$ individually. A consistent low bias in US background $O_3$ in spring was identified, though the specific source of this low bias could not be identified. Here, we extend the bias correction method to estimate biases in separate components of US background $O_3$. Separating the US background $O_3$ components provides new insights into the inferred CTM error in US background $O_3$ that was not possible when US background $O_3$ was treated as a lumped quantity.

## 2 Methods

### 2.1 Chemical transport model simulations

Total $O_3$ (i.e., base $O_3$), US background $O_3$, and individual US background $O_3$ components are simulated at both regional and hemispheric scales using the Community Multiscale Air Quality (CMAQ) model. We use maximum daily 8-h average (MDA8) $O_3$ as the metric of interest since this is the metric used in determining attainment of the NAAQS. References to $O_3$ throughout are to MDA8 $O_3$. CMAQ results are from two recent sets of simulations by the US EPA (Table 1). The two sets of simulations include different US background $O_3$ components allowing us to explore how different components of US background $O_3$ affect the bias in $O_3$.

**Table 1. Simulation names and descriptions for hemispheric-scale and regional-scale simulations. Table adapted from 2020 $O_3$ Policy Assessment Table 2-1 (US EPA, 2020a).**

| Simulation | Description |
| --- | --- |
| BASE | All emission sectors are included. |

| ZUSA | All US anthropogenic emissions are removed including prescribed fires. [a] |
|------|------|
| ZROW | All anthropogenic emissions outside the US are removed including prescribed fires where possible (ROW = rest of world). [b] |
| ZCANMEX | All anthropogenic emissions from Canada and Mexico are removed including prescribed fires where possible. [b] |
| ZANTH | All anthropogenic emissions globally are removed including prescribed fires. [b] |
| STRAT | Tracer species for $O_3$ injected into the upper troposphere/lower stratosphere based on CMAQ potential vorticity parameterization for stratospheric $O_3$.[c] |

[a] Emissions estimated to be associated with intentionally set fires ("prescribed fires") are grouped with anthropogenic fires.
[b] Only for PA simulations
[c] Only for EQUATES simulations.

The first set of simulations was conducted for the Policy Assessment (PA) for the review of the

$O_3$ NAAQS in 2020 (US EPA, 2020a). These simulations also support the draft PA for the reconsideration of the $O_3$ NAAQS. The PA simulations cover the entire year of 2016 and provide estimates of US anthropogenic and US background $O_3$ as well as natural and international anthropogenic contributions to US background $O_3$. International $O_3$ is also further decomposed to short-range international anthropogenic contributions from Canada and Mexico (Canada+Mexico) and long-range international

contributions from other countries. The PA simulations consist of nested simulations from hemispheric scale (Mathur et al., 2017) at 108 km horizontal resolution to continental scale at 36 km resolution to a finer continental scale at 12 km resolution.

US background $O_3$ components are determined by the zero-out method in which the model is run in the same configuration as the base case but with specified emissions sources removed. The zero-out

method is the most common approach for simulating US background $O_3$, though other approaches such as sensitivity simulations and source tagging techniques have also been previously employed (Jaffe et al., 2018). The zero-out method neglects non-linear interactions between sources which can affect the simulated source contribution (Wu et al., 2009; Dolwick et al., 2015). However, the zero-out method is consistent with the definition of US background $O_3$ as the level of $O_3$ in the absence of US anthropogenic

emissions, while sensitivity or tagging techniques would instead provide an estimate of source contributions to total simulated $O_3$ (including $O_3$ from US anthropogenic sources). US background $O_3$ is

estimated by removing US anthropogenic emissions (ZUSA simulation). US anthropogenic $O_3$ is calculated as base $O_3$ minus US background $O_3$. Natural $O_3$ is estimated by removing all anthropogenic emissions (ZANTH simulation). The non-US anthropogenic $O_3$ contribution is estimated by removing anthropogenic emissions everywhere except the US (ZROW simulation). The international contribution is calculated as base $O_3$ minus $O_3$ from the ZROW simulation. Canada+Mexico $O_3$ is estimated by removing Canada and Mexico anthropogenic emissions (ZCANMEX simulation). The Canada+Mexico $O_3$ contribution is calculated as base $O_3$ minus the $O_3$ from the ZCANMEX simulation. Long-range international $O_3$ is estimated as international $O_3$ minus Canada+Mexico $O_3$. Due to non-linear chemistry, there is some residual anthropogenic contribution to base $O_3$ which is not attributed to US or international emissions. Descriptions of these CMAQ simulations and calculation of $O_3$ components are given in Tables S1 and S2. Further details of the modeling setup are available in the 2020 Policy Assessment (US EPA, 2020a).

The second set of simulations was developed from EPA's Air QUAlity TimE Series (EQUATES) project which spans 2002-2019. Additional simulations using the EQUATES modeling framework were conducted for 2016–2017 to estimate US background $O_3$ and US anthropogenic $O_3$ using the zero-out method. The EQUATES simulations consist of hemispheric scale simulations at 108 km horizontal resolution and nested US continental scale simulations at 12 km horizontal resolution. Descriptions of these CMAQ simulations and calculation of $O_3$ components are given in Table S3. Further details on the model configuration for EQUATES are available from Foley et al. (2020) and Foley et al. (2023). More details on both the PA and EQUATES simulations are summarized in Tables S4 and S5.

The 108 km EQUATES simulations also include an inert tracer species which serves as a proxy for simulated stratospheric $O_3$ contributions. Separate stratospheric $O_3$ contributions were not available from the PA simulations, so the EQUATES simulations provide an opportunity to assess potential biases specific to stratospheric $O_3$ contributions. CMAQ simulates stratospheric $O_3$ using a parameterization based on the relationship between $O_3$ and potential vorticity (PV) in the upper troposphere and lower stratosphere (UTLS) (Xing et al., 2016). The parameterization was developed using 21 years of ozonesonde data from the World Ozone and Ultraviolet Radiation Data Centre and PV data from the Weather Research Forecasting (WRF) model for 1990-2010. In the EQUATES 108 km simulations, the

parameterization is applied to the top model layer only. A PV tracer species tracks $O_3$ injected into the UTLS throughout the rest of the model domain for the hemispheric simulations. The 12 km continental simulations inherit the PV tracer species through lateral boundary conditions from the hemispheric simulations. This tracer is subject to transport and deposition but not chemistry. We refer to the PV tracer concentrations as stratospheric $O_3$ since it relates to the stratospheric influence, but it only partly replicates the impact of stratospheric $O_3$ since it does not undergo chemical losses. The stratospheric $O_3$ tracer does, however, provide a measure of the spatiotemporal variability of stratospheric $O_3$ impacts. We also estimate the contribution to US background $O_3$ from sources other than the stratosphere as US background $O_3$ minus the stratospheric $O_3$ tracer and refer to it as non-stratospheric US background $O_3$. The use of the chemically inert PV tracer to split up stratospheric and non-stratospheric influences on US background $O_3$ introduces uncertainty as the stratospheric $O_3$ component may be unrealistically high, especially in areas and times with more active chemistry.

The modeling configurations of the PA and EQUATES simulations differ in some respects which is expected to lead to some differences in simulated $O_3$, though they do share some of the same configuration options. Both the PA and EQUATES simulations use a 44-layer vertical structure for hemispheric scale applications (at 108 km resolution) and a 35-layer vertical structure for continental (i.e., 36 km and 12 km resolution) applications with a vertical extent from the surface to 50 hPa and a surface layer height of approximately 20 m for both the hemispheric and continental configurations (see Mathur et al. (2017) for more details on these vertical layer structures). CMAQ v5.2.1 was used for the PA simulations while CMAQ v5.3.2 was used for the EQUATES simulations. These were the latest versions of CMAQ at the respective times that each set of simulations were conducted. One potential source of differences is updates to halogen chemistry that were introduced in CMAQ v5.3 (Sarwar et al., 2019). These updates in the EQUATES simulations enhance halogen-mediated $O_3$ losses, which are strongest over the oceans. These losses are most relevant for $O_3$ contributions (natural and anthropogenic) that are transported long distances over oceans. An intercomparison of CMAQ v5.2.1 and CMAQ v5.3.1 (which is not significantly different from CMAQ v5.3.2) showed that the newer version typically had lower $O_3$ compared to the older version, with mean bias ~1 ppb lower in CMAQ v5.3.1 (Appel et al., 2021). Besides the updates to halogen chemistry, there are other differences in the chemical mechanisms used for each

set of simulations. The mechanisms used for the hemispheric simulations were cb6r3_ae6_aq for the PA simulations and cb6r3m_ae7_kmtbr for the EQUATES simulations. The part of the mechanism name labeled cb6r3m indicates additional chemistry relevant in marine environments (the halogen chemistry described above); ae6 and ae7 indicate the version number for chemistry relevant to aerosols; aq and kmtbr indicate different treatments of cloud chemistry. The chemical mechanisms used for continental-scale PA and EQUATES simulations (cb6r3_ae6nvPOA_aq and cb6r3_ae7_aq) also differ in their representation of organic aerosols (Murphy et al., 2017; Pye et al., 2019; Qin et al., 2021; Appel et al., 2021) which could affect $O_3$ concentrations. Different versions of WRF (v3.8 for PA simulations and v4.1.1 for EQUATES simulations) employed may also contribute to differences in $O_3$.

Emission inputs also differ between the PA and EQUATES simulations. Different US anthropogenic emission inventories were used for the simulations. The PA simulations used an early version (sometimes called the "alpha" version) of a 2016 emissions modeling platform developed by the National Emissions Inventory Collaborative (US EPA, 2019b). The EQUATES simulations used an inventory that was developed as part of the broader EQUATES framework to model a long timeseries using consistent methods for emissions estimates (Foley et al., 2023). For emissions in Canada and Mexico, both sets of simulations use emission inventories developed by the respective national governments, though the EQUATES simulations use more recent inventories (as described by Foley et al. (2020)) than the PA simulations (as described by US EPA (2019b)). Both the PA and EQUATES simulations use the Tsinghua University inventory of emissions in China (Zhao et al., 2018). For other countries, both sets of simulations use the Hemispheric Transport of Air Pollution (HTAP) v2.2 inventory (Janssens-Maenhout et al., 2015) with scaling factors derived from the Community Emissions Data System (CEDS) (Hoesly et al., 2018) to account for yearly changes. Differences in the anthropogenic emissions used in the two model configurations are expected to contribute to differences in simulated $O_3$, most notably for the different US anthropogenic emissions since we focus here on $O_3$ in the US.

For hemispheric-scale simulations, biogenic VOC emissions are from the Model of Emissions of Gases and Aerosols from Nature version 2.1 (MEGAN2.1) (Guenther et al., 2012). The PA simulations additionally replace MEGAN emissions with emissions from the Biogenic Emission Inventory System (BEIS) (Bash et al., 2016) over North America (US EPA, 2019a). The EQUATES MEGAN emissions

are obtained from a compilation by Sindelarova et al. (2014). Soil $NO_x$ emissions for the PA hemispheric simulations are also from MEGAN with replacement by BEIS soil $NO_x$ over North America. Soil $NO_x$ emissions for the hemispheric EQUATES simulations are from a dataset by the Copernicus Atmosphere Monitoring Service (CAMS, 2018) based on methods by Yienger and Levy (1995). Both the PA and

190 EQUATES simulations use BEIS for biogenic VOC and soil $NO_x$ emissions in the continental-scale simulations. Lightning $NO_x$ emissions for both the PA and EQUATES hemispheric simulations are from monthly climatology obtained from the Global Emissions Initiative (GEIA) and are based on Price et al. (1997). Lightning $NO_x$ was not included in the PA continental-scale simulations, while lightning $NO_x$ for the EQUATES continental-scale simulations is calculated using an inline module in CMAQ (Kang et al.,

2019). For both PA and EQUATES, wildfire emissions outside of North America are based on the Fire Inventory from NCAR (FINN) v1.5 (Wiedinmyer et al., 2011) which provides day-specific fire emissions. Wildfires are vertically allocated with 25% of emissions distributed to the lowest two layers (~0-45 m), 35% distributed to layers 3-9 (~45-350 m), and the remaining 40% distributed to layers 10-19 (~350-2000 m) as described in the Technical Support Document for northern hemispheric emissions (US EPA,

2019a). Wildfire emissions within North America are based on the Hazard Mapping System (HMS) fire product which provides day-specific fire activity data. Emission processing for North American wildfires is further described in the Technical Support Document for North American emissions (US EPA, 2019b) (applicable to PA simulations) and Foley et al. (2023) (applicable to EQUATES simulations). Although the methods are similar, North American wildfire emissions may differ between PA and EQUATES based

on the specific fire activity data that was used in each case. Fire plume injection height for North American fires is determined by an inline plume rise algorithm in CMAQ based on fire heat content (see e.g., Wilkins et al. (2022) for more details on fire plume injection height in CMAQ). Stratospheric $O_3$ in both the PA and EQUATES simulations is from the PV parameterization by Xing et al. (2016) (described in more detail above) in the hemispheric simulations. Stratospheric $O_3$ in the continental-scale simulations

only comes from any stratospheric $O_3$ inherited from the lateral boundary conditions provided by the hemispheric simulations.

## 2.2 O₃ observations

O$_3$ observational data are from the Air Quality System (AQS) database, which provides data from federal, state, local, and tribal air quality monitoring networks across the US. The average precision of O$_3$ monitors in the AQS database was reported as 2.2% and 2.4% in 2016 and 2017, respectively, and the national average absolute bias was reported as 1.5% in both 2016 and 2017 (https://www.epa.gov/amtic/amtic-ambient-air-monitoring-assessments). There were ~360,000 MDA8 O$_3$ observations available per year for 2016 and 2017 from ~1250 unique monitoring sites. These numbers take into account monitoring sites where O$_3$ is measured by multiple instruments at the same location (as indicated in the AQS database by a parameter occurrence code). In these cases, the MDA8 O$_3$ observations from multiple instruments are averaged for a given site and day and treated as a single observation. The observations overrepresent the eastern US compared to the western US. About 40% of daily MDA8 O$_3$ observations and ~36% of O$_3$ monitoring sites are in the western US (as defined by longitude < -97 °W). Western US sites are also overrepresented by sites in the state of California. About 40% of daily MDA8 O$_3$ observations and ~40% of O$_3$ monitoring sites in the western US are in California. The observations also overrepresent the high O$_3$ season of April – October (Figure 1) since many monitors are only required to be operated during the high O$_3$ season.

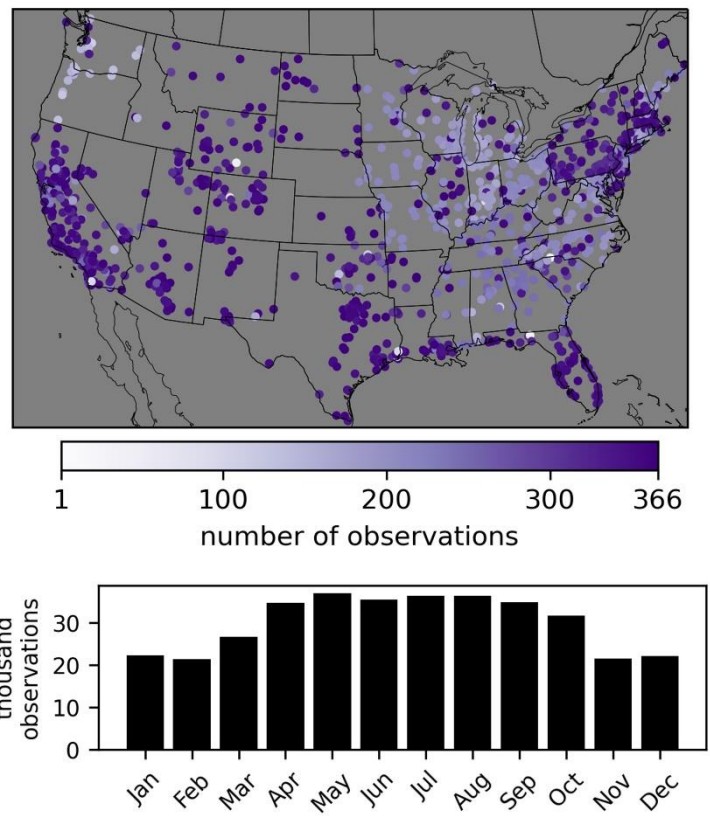

number of observations

**Figure 1. Locations of O₃ observational sites in 2016 indicated with a circle whose color shows the number of daily MDA8 O₃ observations available from each site in 2016 (top). Total number of daily MDA8 O₃ observations in each month of 2016 (bottom).**

### 2.3 O₃ data fusion model

We use multivariate ordinary least squares regression to model the relationship between the individual model components and observed MDA8 O₃. Regression parameters provide estimates of the spatial and temporal model bias attributable to each individual O₃ component. The regression model for ozone mixing ratio O₃ on day *d* and location (*lon*, *lat, z*) is formulated as follows:

$$O_3 = \sum_i \alpha_i O_{3i}^{simulated} + \varepsilon$$

Where:

$$\alpha_i = \alpha_{0,i} + \alpha_{x,i}lon + \alpha_{y,i}lat + \alpha_{z,i}z + \alpha_{sin,i}\sin(d) + \alpha_{cos,i}\cos(d)$$

$d$ is day of year in radians

$z$ is elevation above sea level

*lon*, *lat*, *z*, *sin(d)*, and *cos(d)* are normalized to zero mean and unit standard deviation (Table S6)

$\varepsilon \sim N(0,\sigma^2)$

index $i$ represents different sets of $O_3$ components. Specifically, we consider four sets of $i$:

$i \in \{US\ Anthropogenic, US\ Background\}$ (PA and EQUATES)

$i \in \{US\ Anthropogenic, Natural, International\}$ (PA)

$i \in \{US\ Anthropogenic, Natural, Long\ range\ international, Canada + Mexico\}$ (PA)

$i \in \{US\ Anthropogenic,\ stratospheric\ US\ Background, Stratospheric\}$ (EQUATES)

Each simulated $O_3$ component ($O_{3i}^{simulated}$) is multiplied by the alpha adjustment factor for that component ($\alpha_i$), which varies as a function of space and time, to calculate an adjusted estimate of each $O_3$ component. The inferred model bias for a particular component is calculated as the difference between the original simulated $O_3$ and adjusted $O_3$ for that component. The individual adjusted $O_3$ components are summed to calculate the total adjusted $O_3$. The longitude and latitude terms of $\alpha_i$ are intended to capture the spatial variability of $O_3$ biases while the z term of $\alpha_i$ is intended to capture biases in $O_3$ related to elevation. The sinusoidal day of year terms of $\alpha_i$ are intended to capture the cyclical nature of $O_3$ production and to identify any seasonal dependence in $O_3$ biases. The modeled $O_3$ components do not add up to observed $O_3$ because of biases in the model or its inputs. The CMAQ-simulated $O_3$ components are adjusted by applying estimated regression coefficients to the gridded data so that the sum of the components more closely aligns with observed $O_3$. A more complex method (e.g., nonlinear regression or machine learning) may give a better fit to observed $O_3$, but the interest here is to estimate potential biases in the modeled $O_3$ components which is more straightforward with a linear regression. Empirical orthogonal function (EOF) analysis was used to further explore the spatial and temporal structure of the inferred bias fields and is discussed in the SI.

A separate regression model is developed for each separate model configuration (i.e., model resolution, PA or EQUATES simulation, and US background $O_3$ component split). There are three model

resolutions and three US background $O_3$ splits for the PA simulations, resulting in nine PA models. There are two model resolutions for the EQUATES simulations. The 12 km EQUATES data has two US background $O_3$ splits while the 108 km EQUATES data has one US background $O_3$ split, resulting in three EQUATES models. For the PA models, only 2016 PA simulation data are used to train the models since these simulations are for only that year. For the EQUATES models, both 2016 and 2017 EQUATES simulation data are used to train the models. The location and sampling schedule of the monitoring sites overrepresent the eastern US, low elevations, and high $O_3$ season which may impact how representative the results are for non-monitored locations. Overfitting of the regression model is tested using three cross-validation approaches in which the data are split in both space and time, in space only, and in time only. In the first approach (spatial and temporal withholding), 10% of all observational data are randomly selected and reserved as a test set while the remaining 90% are used as the training set. In the second approach (spatial withholding), data from 10% of randomly selected observation sites are used as a test set while data from the remaining 90% of sites is used as the training set. In the third approach (temporal withholding), data from 10% of randomly selected days of the year are used as a test set while data from the remaining 90% of days of the year are used as the training set. The root mean square error (RMSE) and mean bias for the test and training set are compared to evaluate the potential for the model to overfit the data.

## 3 Results and discussion

### 3.1 CTM results

The overall performance of MDA8 $O_3$ for each simulation is summarized here by the normalized mean bias (NMB) compared to $O_3$ monitoring sites. The 12 km PA simulations were biased high for 2016 (NMB=1.2%) while the 12 km EQUATES simulations were biased low for 2016 and 2017 (NMB=-3.7% and -5.1%). The 36 km and 108 km PA simulations were biased high over the US for 2016 (NMB=5.2% and 10.0%). The 108 km EQUATES simulations were also biased high over the US for 2016 and 2017 (NMB=2.8% and 0.5%). The two sets of simulations are broadly consistent with one another for base, US

anthropogenic, and total US background O$_3$ which are common to both. Details on the contributions from the different O$_3$ components in the PA and EQUATES simulations follow.

CMAQ-simulated annual average MDA8 O$_3$ from the PA simulations show similar results across the three different model resolutions for US background O$_3$ sources (Figure 2; Table 2). Simulated US anthropogenic O$_3$ tends to increase with coarser model resolution which results in corresponding increases in base O$_3$. Natural O$_3$ makes the largest contribution to annual average O$_3$ across the US with a larger contribution in the western US (~55% of base) than in the eastern US (~45% of base). US anthropogenic O$_3$ is the second largest component of annual average O$_3$ with a larger contribution in the eastern US (~35% of base) than in the western US (~20% of base). There are a small number of US grid cells with negative annual averages for US anthropogenic O$_3$. This means that US background O$_3$ was greater than base O$_3$ and indicates that anthropogenic emissions suppress O$_3$ through NO$_x$ titration. Long-range international sources impact the western US (~15% of base) more strongly than the eastern US (~10% of base). Both natural and long-range international O$_3$ tend to be higher at higher elevations, suggesting that some of the effects from natural and long-range international are from O$_3$ in the free troposphere. In spring, O$_3$ lifetimes are longer, and trans-Pacific transport of O$_3$ is more likely which is consistent with the spring peak in long-range international O$_3$ (Liu et al., 1987). The other components and base O$_3$ peak in the summer with some exceptions (Figure 3). In the southeastern US, natural O$_3$ is lower during summer compared to surrounding areas and is lower than natural O$_3$ in the southeastern US during spring. This is likely because O$_3$ loss through reaction with biogenic VOCs (which peak in the summer and are abundant in the southeastern US) reduces O$_3$ under the extremely low NO$_x$ conditions with zero anthropogenic emissions. The Canada+Mexico contribution to O$_3$ is small except at some locations along the border with Mexico where the contributions can be high, especially in the summer. For US grid cells within 100 km of the border with Canada, the annual average impact is ~2 ppb while for US grid cells within 100 km of the border with Mexico, the annual average impact is ~5 ppb.

**Table 2. Summary of annual average of MDA8 O$_3$ components for the Policy Assessment set of simulations. Averages are shown for all of the US and separately for the eastern and western US with a longitude of 97 °W serving as the east-west dividing line. The mean across all grid cells within the given area is shown along with the minimum and maximum for any grid cell within the given**

**area in parentheses. Numbers in the table are in units of ppb. Seasonal averages are provided in Table S13.**

| | Base | US anthropogenic | Natural | Long-range international | Canada+Mexico |
|---|---|---|---|---|---|
| **PA 12 km** | | | | | |
| all US | 39 (18, 56) | 10 (-12, 23) | 20 (15, 30) | 6 (4, 10) | 2 (-4, 9) |
| eastern US | 39 (28, 49) | 13 (2, 23) | 18 (15, 21) | 4 (4, 9) | 1 (1, 6) |
| western US | 40 (18, 56) | 7 (-12, 23) | 22 (15, 30) | 7 (4, 10) | 2 (-4, 9) |
| **PA 36 km** | | | | | |
| all US | 40 (28, 62) | 11 (2, 30) | 20 (15, 28) | 6 (4, 10) | 2 (1, 16) |
| eastern US | 40 (28, 55) | 14 (4, 28) | 18 (15, 21) | 4 (4, 9) | 1 (1, 5) |
| western US | 40 (30, 62) | 8 (2, 30) | 22 (15, 28) | 7 (4, 10) | 2 (1, 16) |
| **PA 108 km** | | | | | |
| all US | 42 (30, 70) | 11 (3, 42) | 21 (16, 28) | 5 (3, 10) | 2 (1, 9) |
| eastern US | 42 (30, 70) | 15 (4, 42) | 19 (16, 23) | 4 (3, 6) | 1 (1, 4) |
| western US | 42 (31, 54) | 8 (3, 20) | 23 (16, 28) | 6 (3, 10) | 2 (1, 9) |

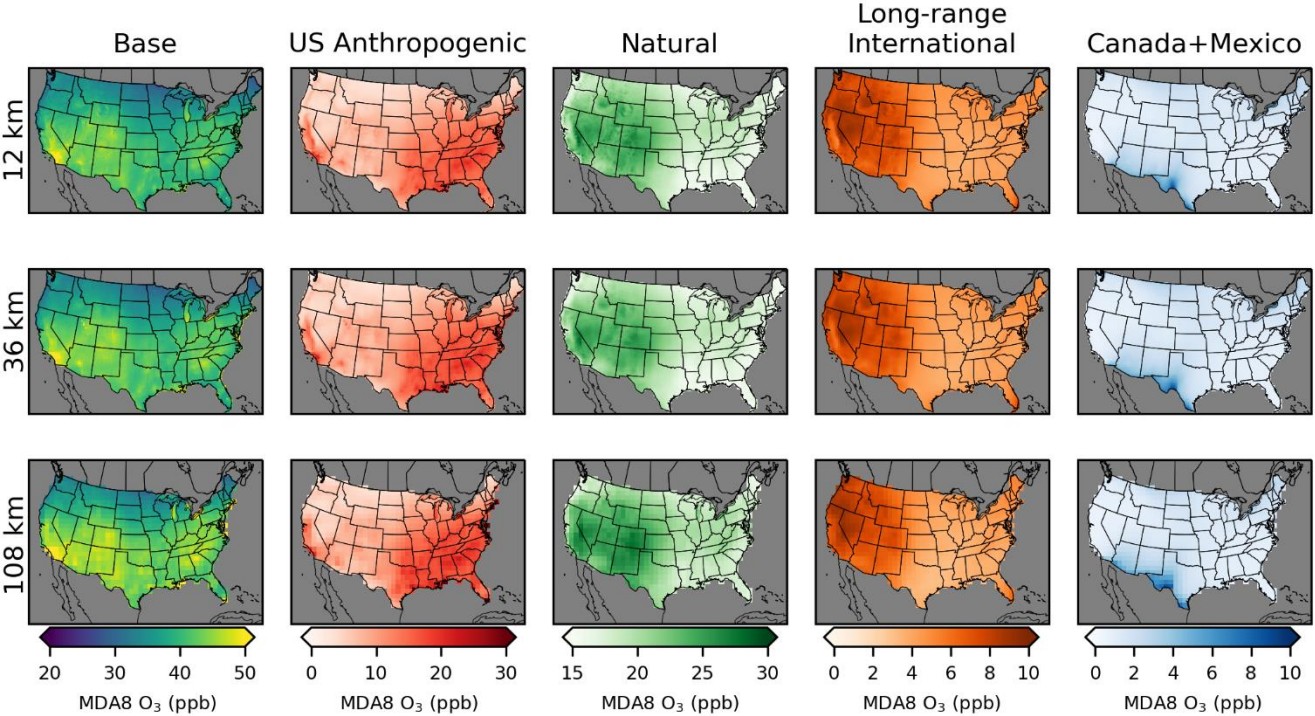

**Figure 2. Annual average MDA8 O₃ from Policy Assessment CMAQ simulations. Results are shown for 12 km (top row), 36 km (middle row), and 108 km (bottom row) horizontal resolutions. O₃ concentrations include total (base) O₃ as well as O₃ components from US anthropogenic, natural, long-range international, and Canada+Mexico sources.**

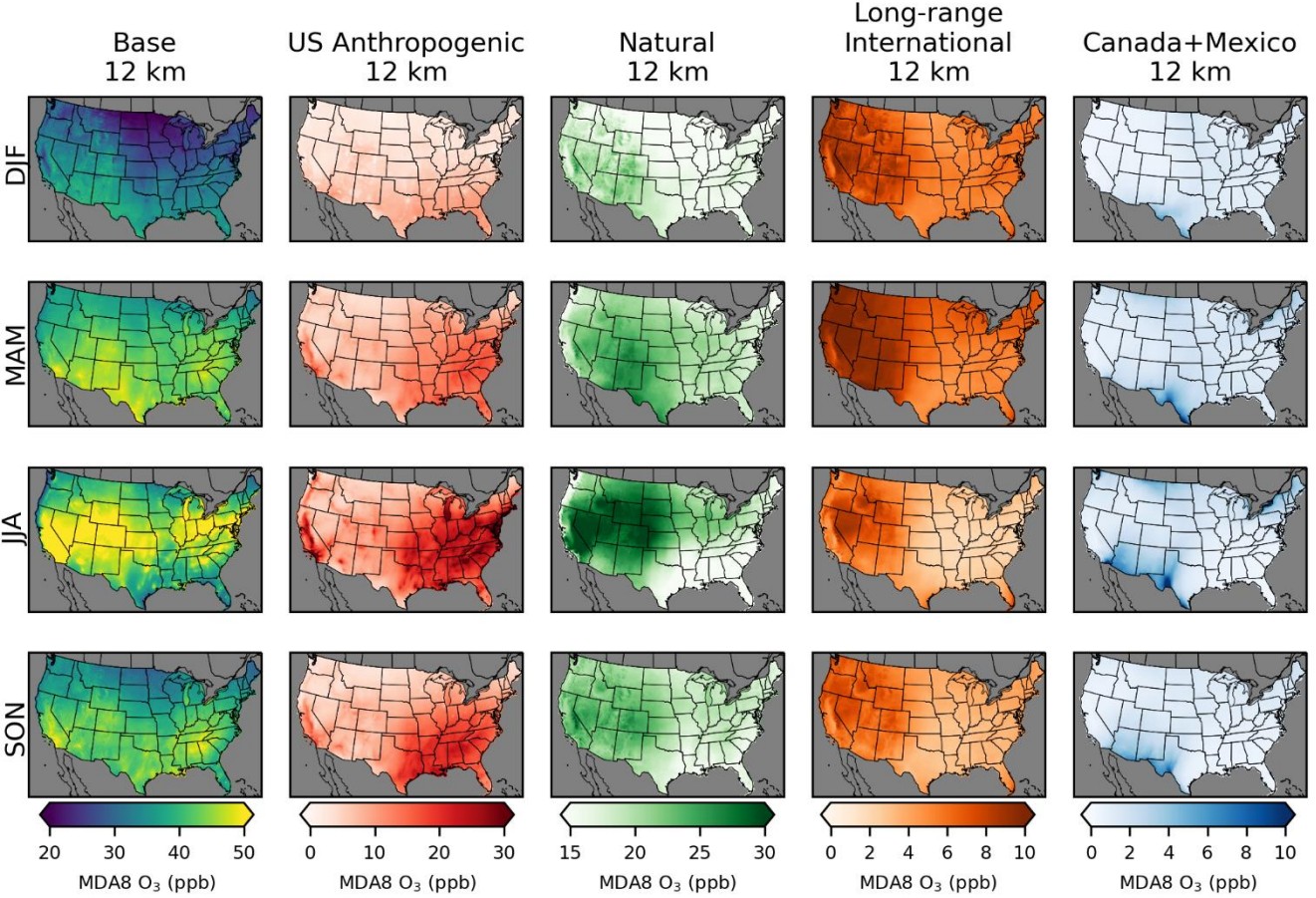

**Figure 3. Seasonal average MDA8 O₃ from Policy Assessment CMAQ simulations. Results are shown for 12 km horizontal resolution for winter (DJF), spring (MAM), summer (JJA), and fall (SON). Seasonal averages for the 36 km and 108 km simulations are provided in Figures S1 and S2. O₃ concentrations include total (base) O₃ as well as O₃ components from US anthropogenic, natural, long-range international, and Canada+Mexico sources.**

While the annual (Figure 2, Table 2) and seasonal (Figure 3) average MDA8 $O_3$ contributions are instructive about longer term contributions, compliance with the NAAQS is determined based on the fourth highest observed MDA8 $O_3$, averaged over three years. We examine the fourth highest total (base) MDA8 $O_3$ along with the contribution from each of MDA8 $O_3$ components on the same day (Figure 4). The areas with the greatest fourth highest MDA8 $O_3$ in the base simulation mostly have large contributions from the US anthropogenic $O_3$ component. This includes much of California and major metropolitan areas in the rest of the US. The eastern US has a higher level of US anthropogenic $O_3$ outside of the metropolitan areas compared to most of the western US where US anthropogenic $O_3$ outside of urban areas is typically in the range of 5-20 ppb. Although the western US and eastern US have similar fourth highest MDA8 $O_3$ values for base $O_3$ (western US 60 ppb, eastern US 61 ppb for 12 km simulations), the western US has a lower average contribution from the US anthropogenic component (14 ppb) compared to the eastern US (33 ppb).

The contribution to the fourth highest MDA8 $O_3$ from natural $O_3$ is largest in parts of the western US with extreme wildfire effects. Large impacts on natural $O_3$ from wildfire events can be seen in Idaho, Wyoming, and California. The contribution from natural $O_3$ is nearly always less than the contribution from US anthropogenic $O_3$ in the eastern US. However, in much of the western US (excluding California and large urban areas) the contribution from natural $O_3$ typically exceeds that of US anthropogenic $O_3$. On average the natural contribution is higher in the western US than in the eastern US (western US 34 ppb, eastern US 22 ppb for 12 km simulations) which reflects the greater prevalence of wildfires in the western US, a larger background contribution from stratospheric $O_3$ due to the higher elevation of the western US, and a larger impact from both long-range and short-range (Canada+Mexico) international sources. The contribution from long-range international MDA8 $O_3$ has a maximum of 20 ppb in the western US and is typically lower in the eastern US compared to the western US on average (western US 6 ppb, eastern US 2 ppb for 12 km simulations). The seasonal average of the long-range international contribution is highest in the spring while base MDA8 $O_3$ is typically highest in the summer (Figure 3), so days with the highest total $O_3$ tend not to be the same days with the highest long-range international $O_3$. The contribution from Canada+Mexico MDA8 $O_3$ is largest in states along the southern and northern borders as expected. Contributions from Canada+Mexico tend to be small except in border areas. The

average MDA8 O$_3$ contributions on days of the top ten highest base MDA8 O$_3$ levels are similar to the results for the fourth highest MDA8 O$_3$ shown here (Figure S3).

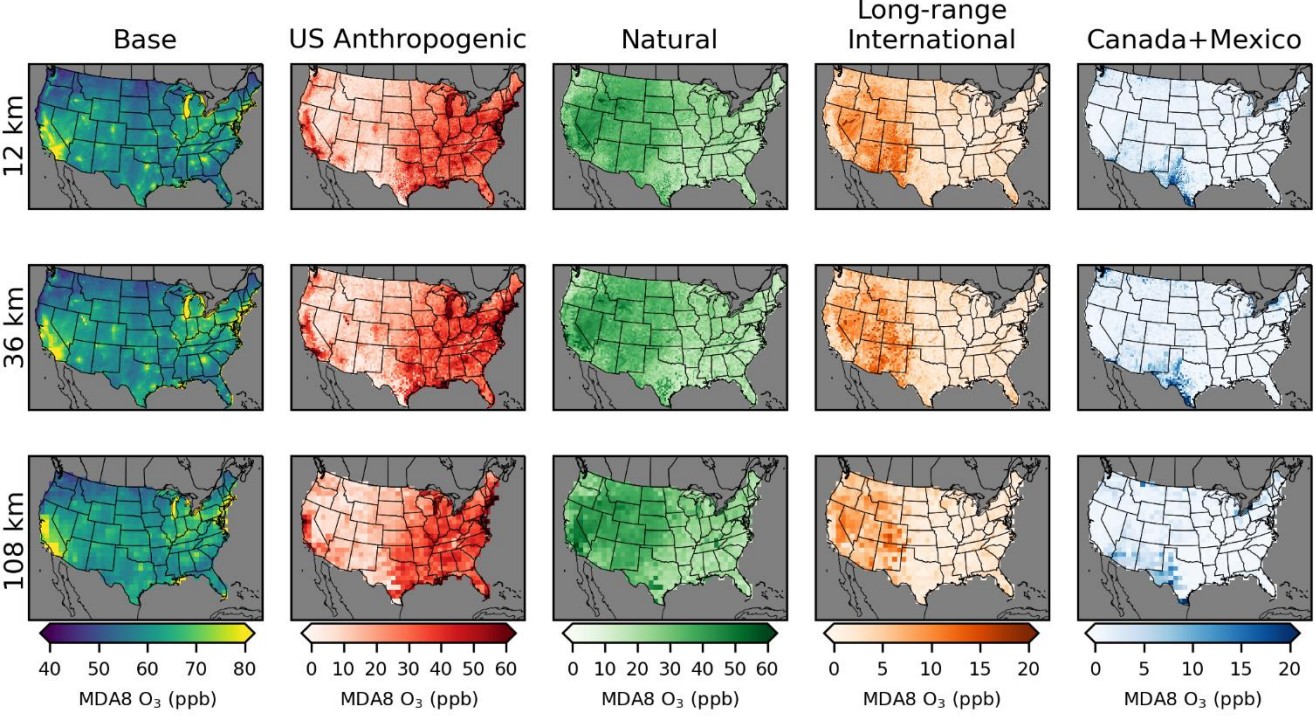

**Figure 4. MDA8 O$_3$ on the day of the fourth highest base case MDA8 O$_3$ from Policy Assessment CMAQ simulations. Results are shown for 12 km (top row), 36 km (middle row), and 108 km (bottom row) horizontal resolutions. O$_3$ concentrations include total (base) O$_3$ as well as O$_3$ components from US anthropogenic, natural, long-range international, and Canada+Mexico**
**sources.**

A second set of simulations (EQUATES) split US background O$_3$ to different components compared to the PA simulations. The use of different US background O$_3$ components provides additional insight into the source-specific biases in US background O$_3$. CMAQ simulated O$_3$ results from the 2016
EQUATES simulations are comparable to the results from the PA simulations for the 12 km simulations, though the EQUATES simulations have slightly less O$_3$ from US anthropogenic and more from US background compared to the PA simulations (Figure 5; Table 3). US anthropogenic O$_3$ contributed ~20%

of annual average base $O_3$ across all US model grid cells (~25% for PA simulations). Like in the PA simulations, the contribution to US anthropogenic $O_3$ was higher in the eastern US (~25% of base) than in the western US (~15% of base). Stratospheric $O_3$ is higher in the western US, especially at higher elevations, which is consistent with previous studies (Jaffe et al., 2018). On average, stratospheric $O_3$ is 40% of base $O_3$ in the western US and 34% of base $O_3$ in the eastern US. Stratospheric $O_3$ represents an upper bound of stratospheric influences because the tracer species used for its calculation in this study does not undergo chemical losses. Non-stratospheric US background $O_3$ contributes 47% of annual average base $O_3$ in the western US and 42% in the eastern US. Non-stratospheric US background $O_3$ is likely underestimated in regions and seasons with more active chemistry due to the use of the chemically inert tracer species used to calculate non-stratospheric US background. The 108 km hemispheric CMAQ (H-CMAQ) results for the EQUATES and PA simulations are similar on average but do have some notable differences. The H-CMAQ simulations are similar in their simulation of US background $O_3$. The US anthropogenic $O_3$ contributions are also similar on average, though the PA simulations have higher maximum values compared to the EQUATES simulations which leads to higher maximum values of base $O_3$.

**Table 3. Summary of annual average of MDA8 $O_3$ components for the EQUATES set of simulations. Averages are shown for all of the US and separately for the eastern and western US with a longitude of 97 °W serving as the east-west dividing line. The mean across all grid cells within the given area is shown along with the minimum and maximum for any grid cell within the given area in parentheses. Numbers in the table are in units of ppb. Seasonal averages are provided in Table S14.**

| | Base | US anthropogenic | US background | Non-stratospheric US background | Stratospheric |
|---|---|---|---|---|---|
| **EQUATES 12 km** | | | | | |
| all US | 39 (22, 51) | 7 (-4, 18) | 32 (24, 44) | 17 (8, 23) | 15 (12, 22) |
| eastern US | 38 (30, 45) | 9 (1, 15) | 29 (24, 36) | 16 (8, 23) | 13 (12, 19) |
| western US | 40 (22, 51) | 5 (-4, 18) | 35 (25, 44) | 19 (12, 22) | 16 (12, 22) |
| **EQUATES 108 km** | | | | | |
| all US | 41 (31, 49) | 8 (2, 18) | 33 (26, 41) | --- | --- |
| eastern US | 40 (31, 49) | 10 (3, 18) | 30 (26, 38) | --- | --- |
| western US | 41 (32, 49) | 6 (2, 12) | 36 (29, 41) | --- | --- |

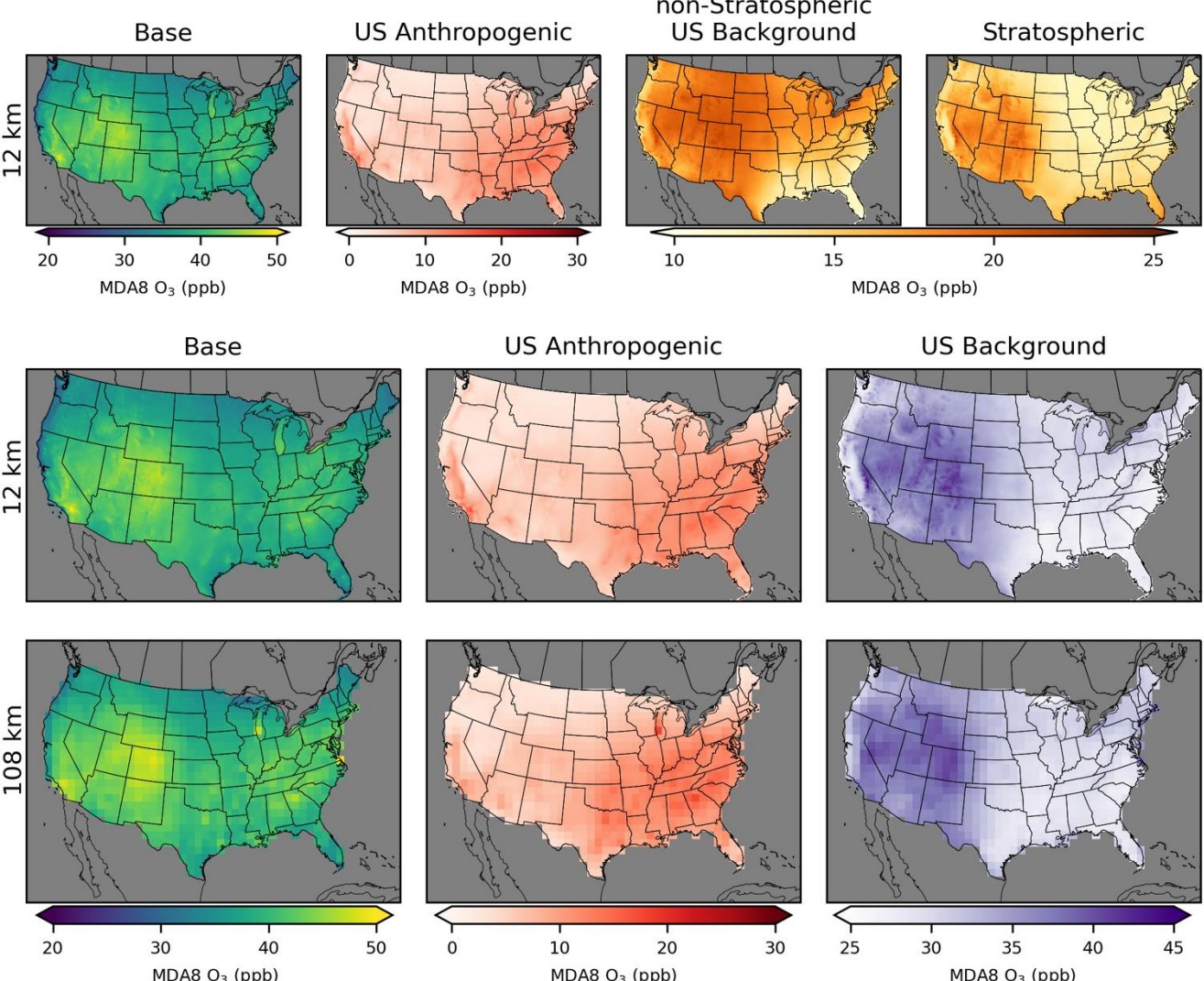

**Figure 5. Annual average MDA8 O$_3$ from EQUATES CMAQ simulations. Results are shown for 12 km resolution (top and middle rows) and 108 km (bottom row). O$_3$ concentrations include total (base) O$_3$ as well as O$_3$ components from US anthropogenic, non-stratospheric US background, and stratospheric sources for 12 km. For both the 12 km and 108 km simulations, O$_3$ concentrations of base, US anthropogenic, and total US background are also shown.**

Base $O_3$ in EQUATES is highest in the summer (Figure 6). US background $O_3$ is the highest during spring throughout most of the US. However, in much of the Mountain West, US background $O_3$ is highest during the summer (Figures S4 and S5). The stratospheric $O_3$ tracer is the highest in the western US. Much of the western US has stratospheric $O_3$ at about the same level in the spring and summer. In the southeastern US, stratospheric $O_3$ is highest in the summer while in the northeastern US, there are similar levels of stratospheric $O_3$ in the spring and summer. Stratospheric $O_3$ is elevated in the summer because of the lack of chemical sinks due to the inert tracer species used to estimate stratospheric $O_3$. Most previous studies have indicated that stratospheric $O_3$ peaks in the spring (Lin et al., 2015). The stratospheric contribution to $O_3$ from H-CMAQ calculated using the decoupled direct method (which does account for chemical losses) also showed higher stratospheric contributions in spring than in summer (Mathur et al., 2022). The higher summer stratospheric $O_3$ here is explained by the lack of chemical losses due to the tracer method used. Potential biases are explored further in Section 3.3. US anthropogenic $O_3$ is highest in the summer in the eastern US and in California, consistent with the PA simulations. Non-stratospheric US background $O_3$ is relatively uniform outside of summer, though it tends to be slightly lower in the southeast and higher in the western US.

The results from both the PA and EQUATES simulations indicate that US background $O_3$ contributes more than US anthropogenic $O_3$ to base $O_3$ on an annual average basis. Simulated US background $O_3$ is higher in the western US than in the eastern US due to greater impacts from both natural and non-domestic anthropogenic sources. Simulated US anthropogenic $O_3$ is higher in the eastern US than in the western US due to the higher population density and consequently greater anthropogenic emissions. The contributions from US anthropogenic $O_3$ peak in the summer which causes base $O_3$ to peak in the summer as well. US background $O_3$ varies by season but is not as seasonally variable as US anthropogenic $O_3$. These results are broadly consistent with previous efforts to quantify US background and US anthropogenic $O_3$ using CTMs (Mcdonald-Buller et al., 2011; Jaffe et al., 2018).

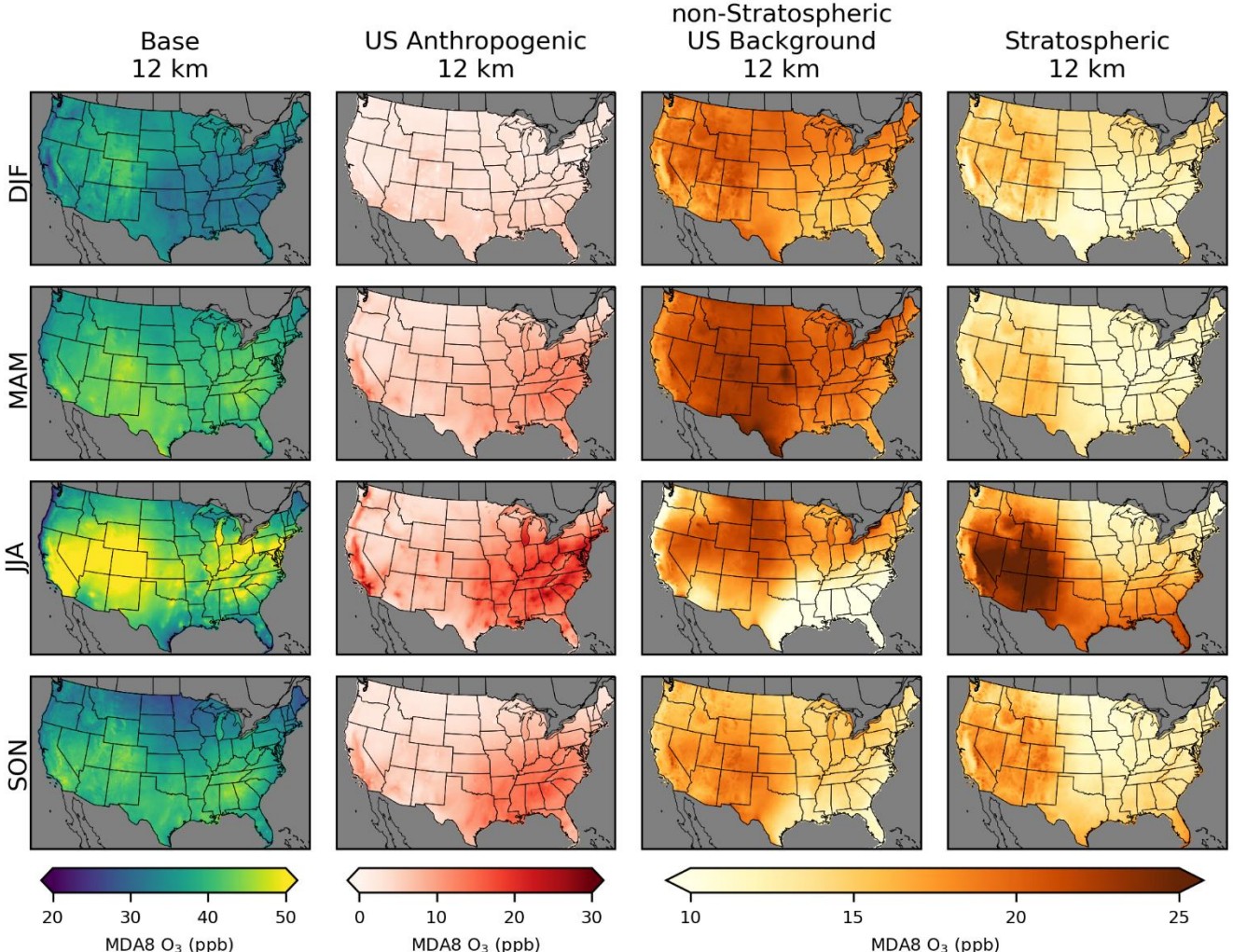

**Figure 6. Seasonal average MDA8 O₃ from EQUATES CMAQ simulations. Results are shown for 12 km horizontal resolution for winter (DJF), spring (MAM), summer (JJA), and fall (SON). O₃ concentrations include total (base) O₃ as well as O₃ components from US anthropogenic, non-stratospheric US background, and stratospheric sources. Seasonal averages for the other US background O₃ split cases are provided in the SI (Figures S4 and S5).**

Similarly to the PA simulations, we examine the fourth highest total (base) MDA8 O₃ along with the contribution from each of MDA8 O₃ components on the same day for the EQUATES simulations. Like in the PA simulations, the areas with the greatest fourth highest MDA8 O₃ values for base MDA8

$O_3$ tend to have larger contribution from US anthropogenic $O_3$ than from US background $O_3$. The EQUATES fourth highest base MDA8 $O_3$ is slightly lower than in the PA simulations (56 ppb in the western and 57 ppb in the eastern US compared to 60 and 61 ppb in the PA simulations at 12 km). The US anthropogenic contribution is similarly lower in the EQUATES simulations (10 ppb in the western US, 25 ppb in the eastern US compared to 14 and 33 ppb in the PA simulations at 12 km). The contributions from US background $O_3$ are higher in the western US than in the eastern US on average (eastern US 32 ppb, western US 46 ppb for 12 km simulations). The contribution from non-stratospheric US background $O_3$ (western US 25 ppb, eastern US 18 ppb) is generally greater than the contribution from stratospheric US background $O_3$ (western US 21 ppb, eastern US 14 ppb). The western US has larger contributions from stratospheric $O_3$, long-range international $O_3$, and wildfires. In the EQUATES simulations, the Flint Hills area of Kansas stands out as an area influenced by fires. The fires in this area are typically prescribed burning of grasslands used for agricultural land management. While these were included in the fire emissions for the US background $O_3$ simulation, prescribed burns are typically classified as anthropogenic sources rather than background sources. The average MDA8 $O_3$ contributions on days of the top ten highest base MDA8 $O_3$ levels are similar to the results for the fourth highest MDA8 $O_3$ shown here (Figure S6).

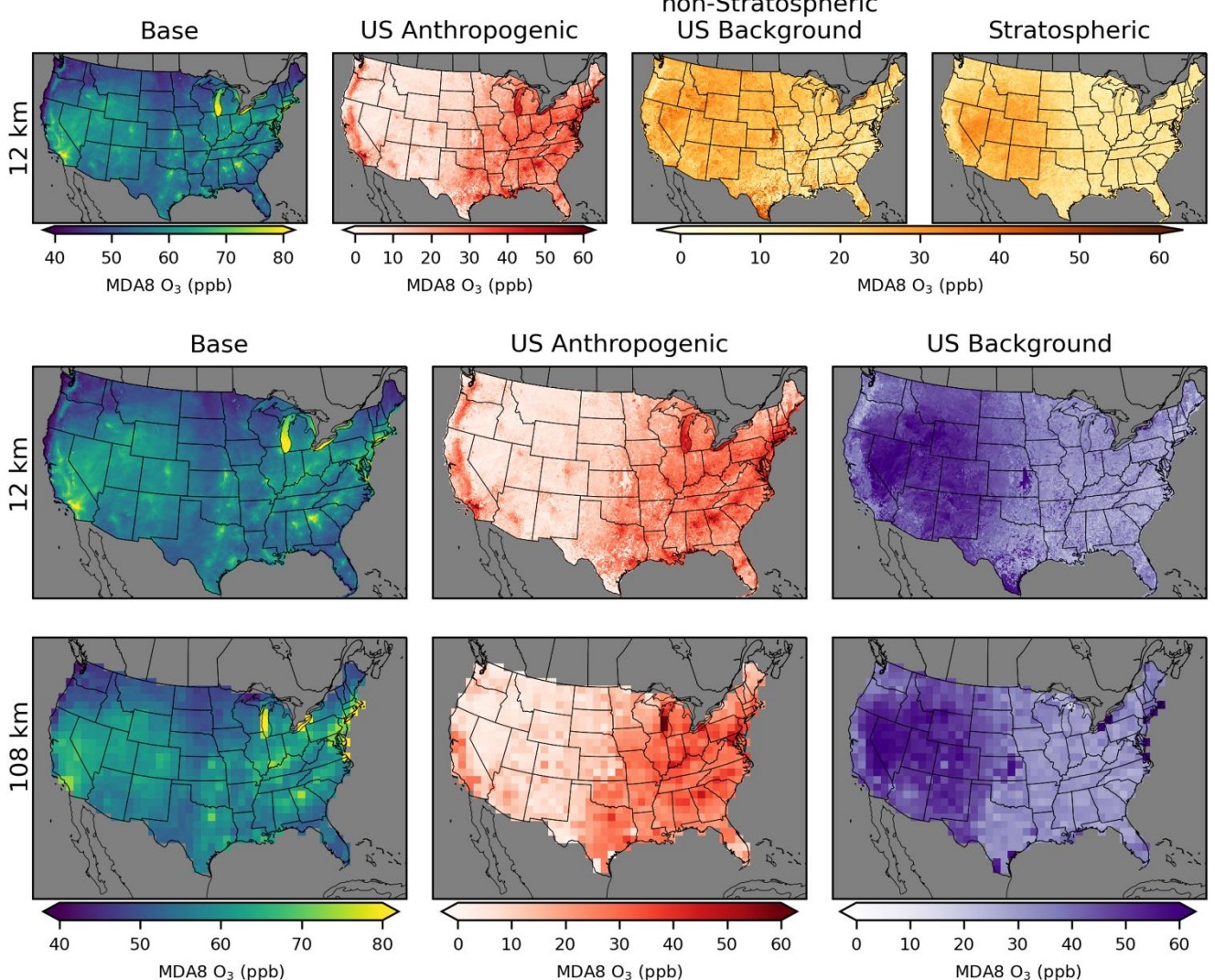

**Figure 7. MDA8 O₃ on the day of the fourth highest base case MDA8 O₃ from EQUATES CMAQ simulations. Results are shown for 12 km resolution (top and middle rows) and 108 km (bottom row). O₃ concentrations include total (base) O₃ as well as O₃ components from US anthropogenic, non-stratospheric US background, and stratospheric sources for 12 km. For both the 12 km and 108 km simulations, O₃ concentrations of base, US anthropogenic, and total US background are also shown.**

## 3.2 Cross-validation of regression modeling

Overfitting is tested using a cross-validation analysis as described in Section 2.2. Three different cross-validation methods are used: spatial and temporal withholding, spatial withholding, and temporal withholding. The parameters derived from the training set are then used to predict the observed $O_3$ in the test set. The RMSE and mean bias with respect to the true observations of both the training and test set are compared to one another (Table 4; Tables S7 and S8). For each of the three cross-validation methods,

the RMSE and mean bias of the training and test sets are similar to one another. This indicates that the model is not overfitting and is generalizable to data outside of its training data, providing confidence that we can apply the regression models to the gridded CTM results to estimate the bias in $O_3$ and individual $O_3$ components across the US.

**Table 4. Summary of performance for cross-validation of MDA8 $O_3$ data fusion model. Values shown are the average over all regression model cases. RMSE and mean bias statistics for individual cases are provided in Tables S7 and S8. The performance for the base $O_3$ simulations prior to applying the bias adjustment is also provided for comparison.**

| metric | Base simulations | spatial and temporal withholding | | spatial withholding | | temporal withholding | |
|---|---|---|---|---|---|---|---|
| | | training | test | training | test | training | test |
| RMSE (ppb) | 9.53 | 7.80 | 7.83 | 7.83 | 7.58 | 7.81 | 7.79 |
| mean bias (ppb) | 1.13 | -0.19 | -0.20 | -0.19 | -0.63 | -0.19 | 0.38 |

## 3.3 Inferred CTM biases

The coefficients from the regression models (Tables S9 – S12) are applied to the gridded CTM data to calculate adjusted values of each $O_3$ component. The inferred CMAQ bias for each component is the difference between the original CMAQ-simulated value and the adjusted value. The inferred bias in base $O_3$ is the original CMAQ-simulated base $O_3$ minus the sum of adjusted $O_3$ components. For the PA

simulations, there is a residual anthropogenic component of base $O_3$ that is not apportioned to either US anthropogenic or international sources due to the effects of non-linear chemistry (Table S2). The residual anthropogenic component is equal to base $O_3$ – natural $O_3$ – international $O_3$ – US anthropogenic $O_3$. This means that the sum of biases in the individual components do not add up to the bias in base $O_3$ as the

residual anthropogenic component was not included in the adjusted $O_3$ results. In the PA simulations,
base $O_3$ is inferred to be biased high in most of the Eastern US as well as in some parts of California and
Arizona (Figure 8). US anthropogenic $O_3$ is inferred to be biased high in the same areas. Reducing the
amount of US anthropogenic $O_3$ improves the fit to base $O_3$ which is suggestive that biases in the effects
from US anthropogenic emissions contribute to the high biases inferred in base $O_3$. The inferred high
biases in base and US anthropogenic $O_3$ increase with increasing coarseness of model resolution in the
eastern US. Similarly, the high bias increases with coarser model resolution in the Canada+Mexico
component along the border with Mexico. The inferred high biases in US anthropogenic $O_3$ in the eastern
US are primarily driven by biases in the summer and fall (Table S15, Figures S7-S9). Inferred eastern US
anthropogenic $O_3$ biases average 2, 7, and 11 ppb in the summer and 3, 4, and 5 ppb in the fall for the 12,
36, and 108 km simulations. In the western US, where US anthropogenic $O_3$ is mostly found to be biased
low, coarser model resolution results in the summer average bias changing from slightly negative in the
12 km simulations (-0.5 ppb) to slightly positive in the 36 and 108 km simulations (+0.7 ppb and +1.0
ppb).

In contrast to our results showing an increase in $O_3$ with coarser resolution, Schwantes et al. (2022)
found that $O_3$ tended to increase for a finer resolution simulation (~14 km vs. ~111 km over the CONUS)
during the summer over urban areas using the Community Earth System Model (CESM)/Community
Atmosphere Model with full chemistry (CAM-chem) model which was attributed to improvements in the
spatial resolution of $NO_x$ emissions resulting in less artificial dilution of $NO_x$ and enhanced $O_3$ production.
Similarly, Lin et al. (2024) found that a variable resolution global model (AM4VR with horizontal
resolution of 13 km over the CONUS) had increased $O_3$ over urban areas compared to a fixed resolution
model (AM4.1 with horizontal resolution of ~100 km globally). In particular for the Los Angeles Basin
and Central Valley regions of California, Lin et al. (2024) found that the increased resolution of AM4VR
led to better simulation of observed $O_3$ levels in these areas due the finer resolution model's ability to
represent sharp spatial gradients in areas with $NO_x$-limited vs. $NO_x$-saturated $O_3$ production regimes. Our
analysis of the fourth highest MDA8 $O_3$ levels shows similar findings over California (Figures 4 and 7).
Given the previous results finding increased $O_3$ with finer resolution simulations, our results here finding
higher biases in US anthropogenic $O_3$ in the eastern US with coarser resolution should be taken to apply

specifically to the CMAQ model results described here rather than as a general finding on the impact of model resolution on $O_3$ production. Additionally, given that the finding of higher US anthropogenic $O_3$ with coarser model resolution does not hold for the analysis of the fourth highest MDA8 $O_3$ levels, this finding should be taken to apply only to longer term (e.g., annual or seasonal) averages.

There are offsetting inferred biases in the long-range international and natural $O_3$ components in much of the western US. The offsetting inferred biases may reflect an inability of the regression model to separate the signals from long-range international and stratospheric $O_3$. Long-range international and stratospheric $O_3$ are expected to impact sites at similar spatial and temporal scales, with larger impacts expected at high elevations in the western US during spring. Stratospheric $O_3$ effects are not limited to episodic intrusion events but also come from constant entrainment of stratospheric air to the free troposphere. The impacts from long-range international emissions are primarily from long-range transport in the free troposphere, so stratospheric $O_3$ and long-range international $O_3$ are expected to be correlated. The regression model may be assigning bias due to stratospheric $O_3$ to long-range international $O_3$ because the CTM-modeled long-range international component has more correlation with the stratospheric $O_3$ impact than the CTM-modeled natural component. This could result in the regression model adjusting long-range international $O_3$ upwards (i.e., inferred negative bias) to add stratospheric $O_3$. The natural $O_3$ is then adjusted downwards (i.e., inferred positive bias) in the same locations because some of the effects of stratospheric $O_3$ are captured in the CTM-modeled natural $O_3$ component but need to be offset because of the $O_3$ that was added to the long-range international component. This indicates a limitation of this method in that it is sensitive to correlation between modeled $O_3$ components. Correlation of the $O_3$ components is a major confounding issue in this analysis. In interpreting the results, it is necessary to consider both the inferred biases and the correlation of the components together.

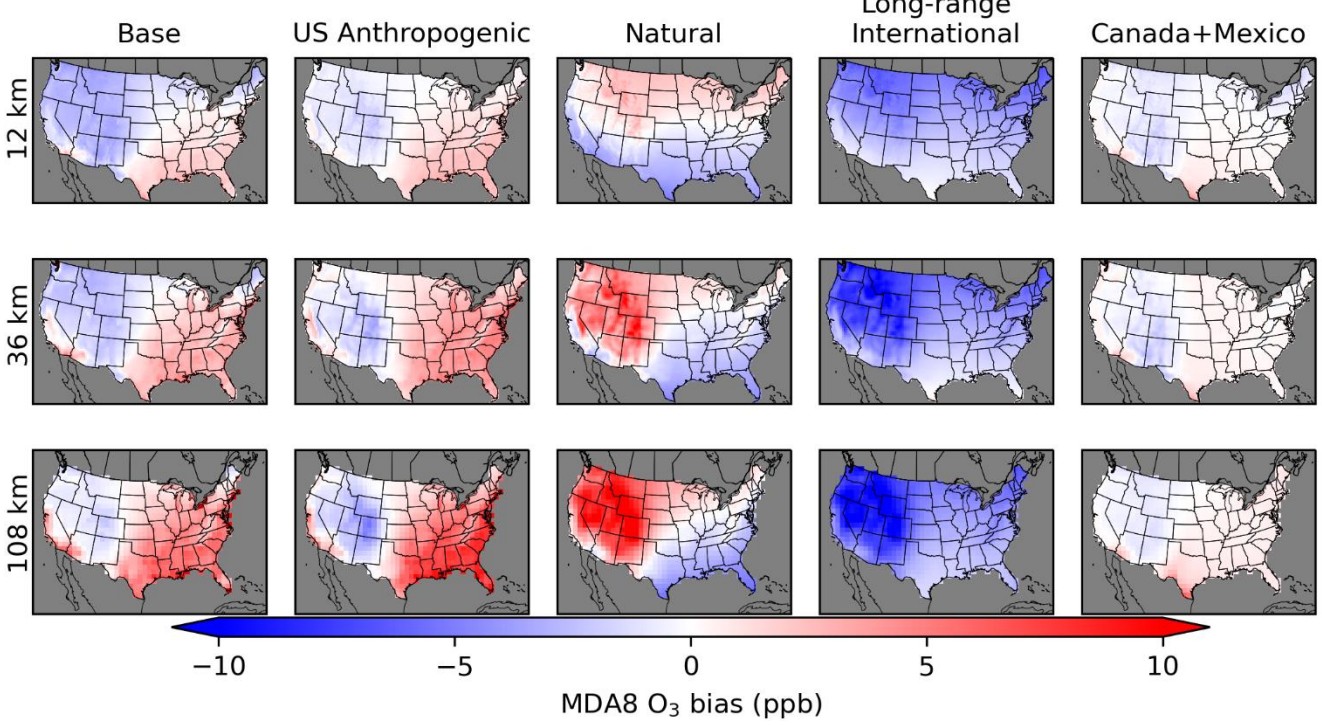

**Figure 8. Annual average of inferred MDA8 O₃ model bias from Policy Assessment CMAQ simulations. Results are shown for 12 km (top row), 36 km (middle row), and 108 km (bottom row) horizontal resolutions. O₃ concentrations include total (base) O₃ as well as O₃ components from US anthropogenic, natural, long-range international, and Canada+Mexico sources. Seasonal averages are provided in Figures S7-S9.**

In the temporal trends of inferred base O₃ bias, the PA simulations show a consistent low bias in winter and spring and high bias in summer and fall which is consistent across model resolution scales (Figure 9). There is also a consistent high bias in US anthropogenic O₃ in summer and fall in the eastern US which increases with coarser model resolution. Inferred bias in US anthropogenic O₃ in the western US has some small seasonal variability but is near zero on average. The seasonal patterns of long-range international O₃ bias have the largest underestimate in the winter and spring and the smallest underestimate in late summer and early fall. The temporal trend of natural O₃ differs in the 12 km simulation compared to the 36 km and 108 km simulations. In the 12 km simulation, natural O₃ biases

are higher in the middle of the year than in the beginning and end of the year. In the 36 km and 108 km simulations, the opposite is found. This change in sign is a result of changes in the spatial patterns of natural $O_3$ inferred bias in different seasons. In the 12 km simulation, natural $O_3$ is inferred to be biased low in the southern part of the US and biased high in the northern part of the US. In the 36 km and 108 km simulations natural $O_3$ is inferred to be biased low in the eastern US and mostly biased high in the

western US, particularly in the Mountain West region. These spatial changes in the seasonal average natural $O_3$ bias are enough to change the sign of the US average temporal bias trend. As described before, the offsetting negative long-range international bias and positive natural $O_3$ bias in the high elevation areas of the western US are thought to be a result of the regression model allocating stratospheric $O_3$ bias to the long-range international $O_3$ signal while removing some stratospheric $O_3$ from the natural $O_3$ signal.

Canada+Mexico $O_3$ biases are very small when averaged across the US since this source primarily affects border areas and only has small impacts elsewhere.

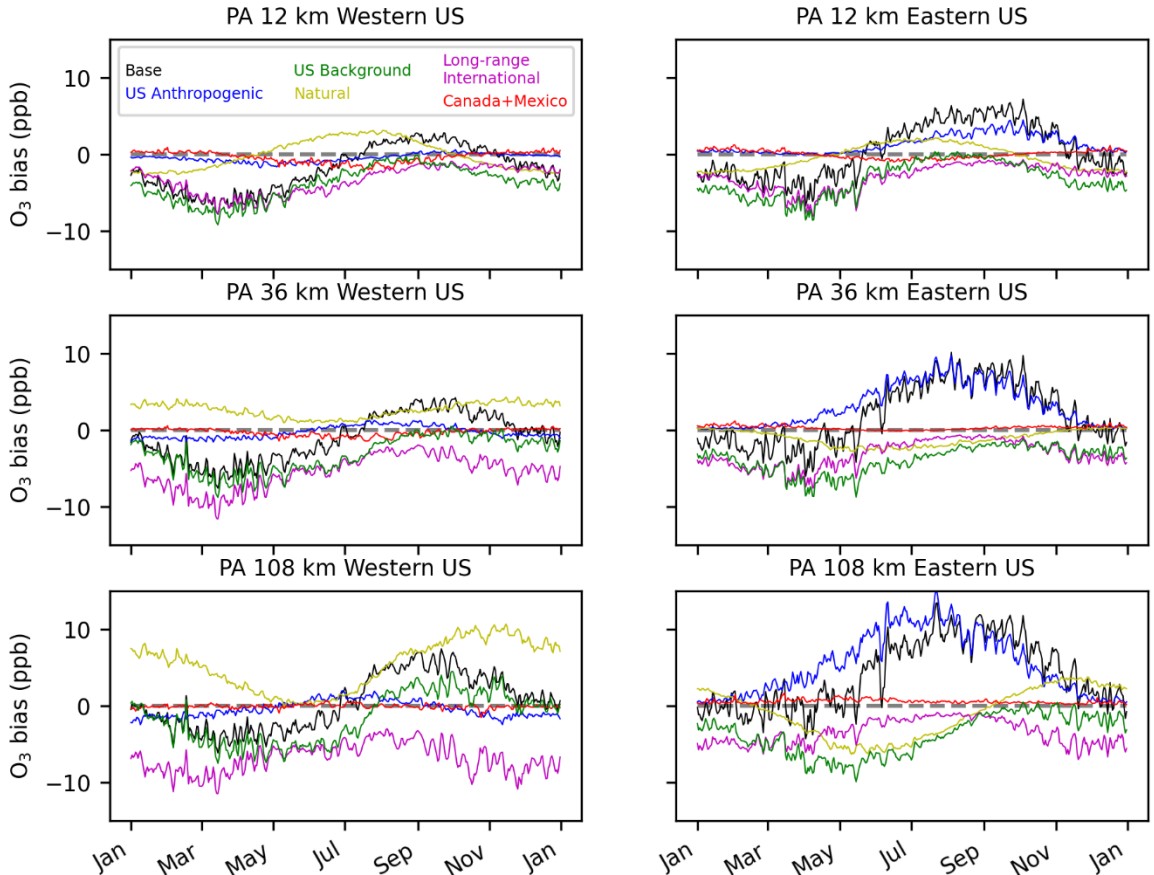

**Figure 9. Daily average of inferred MDA8 O₃ model bias from Policy Assessment CMAQ simulations averaged across US model grid cells in the eastern and western US. A longitude of 97 °W is used as the dividing line between east and west. PA O₃ concentrations include total (base) O₃ as well as O₃ components from US anthropogenic, natural, long-range international, and Canada+Mexico sources. US background indicates the sum of biases for individual US background components.**

The spatial results for the EQUATES 12 km simulations are shown for two O₃ split cases. One case splits US background O₃ to stratospheric and non-stratospheric sources while the other considers all US background O₃ together. Results show a mostly low bias inferred in base O₃ throughout most of the US for the 12 km simulation (Figure 10). For the 108 km H-CMAQ simulation there is a high bias in the eastern US and a low bias in the western US for base O₃. Like the PA results there is a high bias in US

anthropogenic $O_3$ in the eastern US that increases with coarser model resolution. The inferred low bias in the stratospheric $O_3$ component indicates that there is too little stratospheric $O_3$ in the western US. There is an inferred high bias in stratospheric $O_3$ in the eastern US. The stratospheric $O_3$ results should be interpreted with some caution because the stratospheric component comes from a chemically inert tracer.

The stratospheric $O_3$ biases are partly offset by opposite biases in the non-stratospheric US background $O_3$. The low biases in stratospheric $O_3$ and the lack of low biases in the non-stratospheric US background $O_3$ provides more evidence that the low biases in the long-range international $O_3$ from the PA simulations are related to low biases in stratospheric $O_3$.

In the case where US background $O_3$ is not split into stratospheric and non-stratospheric

components, the 12 km and 108 km simulations both have low biases in US background $O_3$, but the magnitude of bias is greater in the 12 km simulation than in the 108 km simulation. This may be a result of differences in the impacts of stratospheric $O_3$ at the surface level in the H-CMAQ simulation compared to the continental-scale simulation. Differences in the estimation of stratospheric $O_3$ impacts may arise from differences in how the vertical structure of the model in the H-CMAQ simulations is configured

compared to the continental simulations. The UTLS PV $O_3$ scaling is turned on during the H-CMAQ simulation. For the continental simulation, PV $O_3$ scaling is turned off because the continental model configuration uses fewer vertical layers and a coarser vertical resolution in the UTLS compared to the H-CMAQ simulations. The stratospheric $O_3$ influences in the continental simulation are only those influences that are inherited from the lateral boundary conditions. Previous work indicates that $O_3$ in the

upper layers of the continental-scale model is driven mostly by horizontal advection of the lateral boundary conditions (Hogrefe et al., 2018), meaning that if stratospheric intrusion events are captured by the hemispheric-scale simulation, the effects of these events are also expected to be captured by the continental-scale simulation. However, a sensitivity test with UTLS PV $O_3$ scaling turned on during the continental simulation may be an area for future study. This would require the addition of more vertical

layers with finer resolution in the UTLS in the continental simulation to support the PV $O_3$ scaling parameterization. The differences in vertical structure of the hemispheric and continental simulations can affect the vertical mixing of stratospheric $O_3$ from upper layers down to the surface which may explain the differences in inferred bias of US background $O_3$. Alternatively, the differences in US background $O_3$

biases could also occur due to differences in $O_3$ production from local US background $O_3$ sources across
model resolution scales and may not necessarily be affected by differences in stratospheric $O_3$.

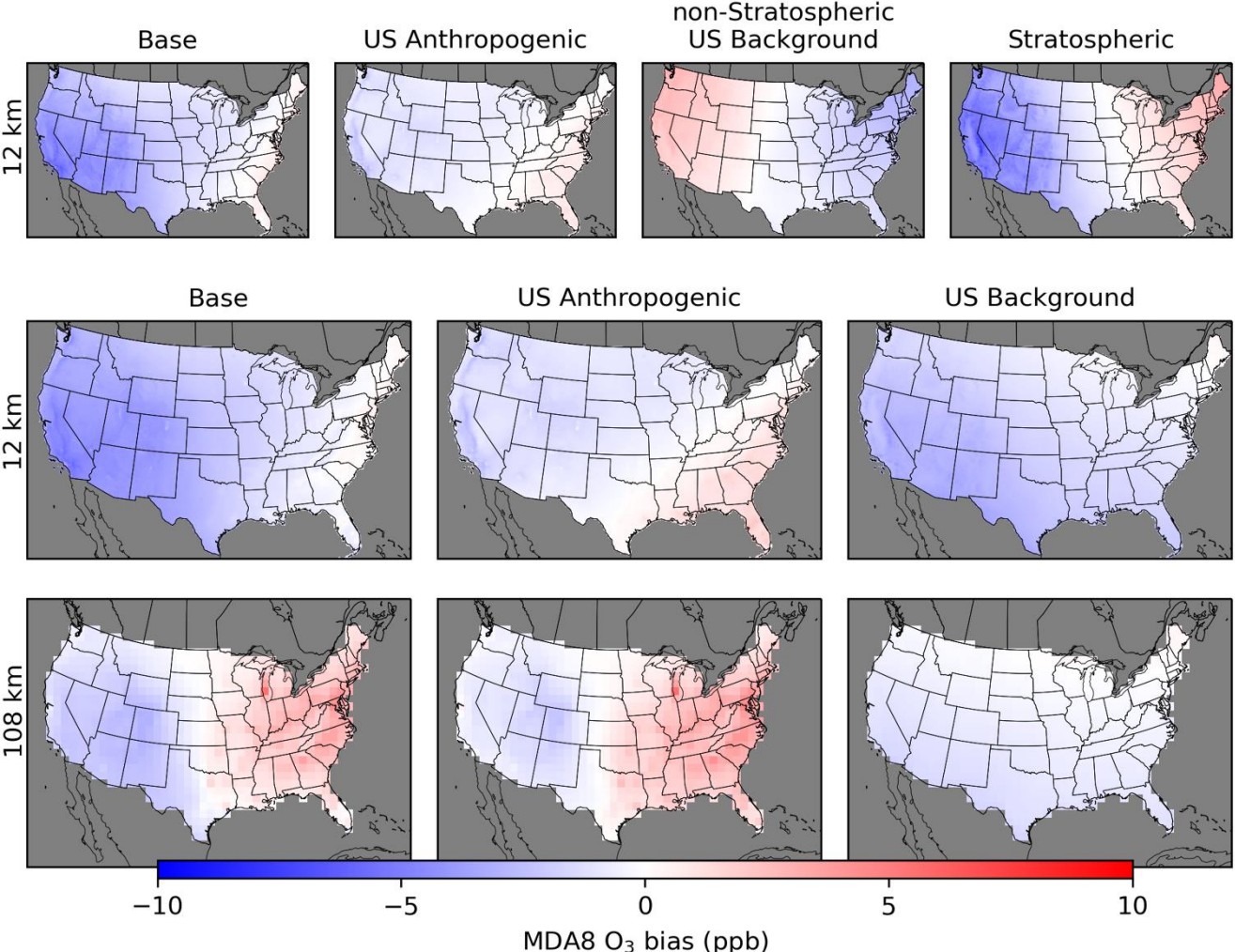

**Figure 10. Annual average of inferred MDA8 $O_3$ model bias from EQUATES CMAQ simulations. Results are shown for 12 km resolution (top and middle rows) and 108 km (bottom row). $O_3$ concentrations include total (base) $O_3$ as well as $O_3$ components from US anthropogenic, non-stratospheric US background, and stratospheric sources for 12 km. For both the 12 km and 108 km simulations, $O_3$ concentrations of base, US anthropogenic, and total US background are also shown. Seasonal averages are provided in Figures S10-S12.**

For the EQUATES temporal results, base $O_3$ is biased low in the spring and high in the summer in the eastern US (Figure 11). In the western US, base $O_3$ is biased low throughout most of the year. Averaged across the US, bias is near zero in the summer and fall in the 12 km simulation with high biases in the 108 km simulation during the same period (+1 ppb in summer; +2 ppb in fall). The high biases in base $O_3$ in the eastern US are mostly due to high biases in the US anthropogenic $O_3$ component which peak in the summer (average +1.4 and +6.0 ppb for the 12 and 108 km simulations) and continue to be biased high into the fall (average +0.8 and +2.2 ppb for the 12 and 108 km simulations). The stratospheric $O_3$ component is inferred to be biased low except in the summer and early fall. In the western US, stratospheric $O_3$ bias in the summer is near zero in the summer and fall while in the eastern US, stratospheric $O_3$ is biased high in the summer and fall. The lowest biases in stratospheric $O_3$ occur in the winter. The stratospheric $O_3$ biases are partially offset by opposing biases in the non-stratospheric US background $O_3$. The regression model formulation without the separate stratospheric $O_3$ indicates that there is a low bias in US background $O_3$ throughout most of the year in the 12 km simulation which is at its lowest in the spring. The 108 km simulations show a low bias for US background $O_3$ in the spring and summer and high bias in the fall and winter.

In the 12 km EQUATES simulations, the stratospheric $O_3$ tracer averages 14 ppb in the western US during spring, with a maximum spring average across all western US grid cells of 17 ppb. Using the bias correction approach developed here, we find that the spring average stratospheric $O_3$ in the western US is biased low by 3.5 ppb, resulting in an adjusted (i.e., bias corrected) estimate of western US spring average stratospheric $O_3$ of 17 ppb. Consistent with the low bias in stratospheric $O_3$ suggested here, other CTMs have estimated higher stratospheric $O_3$ contributions compared to those simulated here with CMAQ. The spring average of stratospheric $O_3$ contributions estimated with the AM3 model has been estimated at 20-25 ppb (Lin et al., 2012a; Langford et al., 2015; Lin et al., 2015). The AM3 estimates of stratospheric $O_3$ have sometimes been estimated to be biased high (Lin et al., 2012a) and have also been shown to lead to overestimated springtime $O_3$ concentrations when used as boundary conditions for regional-scale CMAQ simulations (Hogrefe et al., 2018) but at other times have been estimated to be relatively unbiased based on evaluation against observations from intensive field studies (Langford et al., 2015). The stratospheric $O_3$ contribution simulated by AM3 has been previously found to be higher than

that of the GEOS-Chem global model (Fiore et al., 2014). Using GEOS-Chem, Zhang et al. (2014) found the spring mean stratospheric $O_3$ influence in the Intermountain West to range from 8-10 ppb as estimated using the standard GEOS-Chem definition of stratospheric $O_3$ as described in Zhang et al. (2011) and, alternatively, found a spring mean of 12-18 ppb using a definition of stratospheric $O_3$ adopted from Lin et al. (2012a) (the same method used for the AM3 estimates reported here). Itahashi et al. (2020) previously found that the stratospheric $O_3$ representation in CMAQ was biased low in the free troposphere and suggested that improvements were needed to the CMAQ representation of stratosphere to troposphere transport. Our bias adjusted estimate of western US spring mean stratospheric $O_3$ (17 ppb) falls in between the estimates from the default GEOS-Chem representation (8-10 ppb) and from AM3 (20-25 ppb). As these are seasonal averages, the values are more representative of the continual entrainment of stratospheric air into the troposphere rather than episodic deep stratospheric intrusion events.

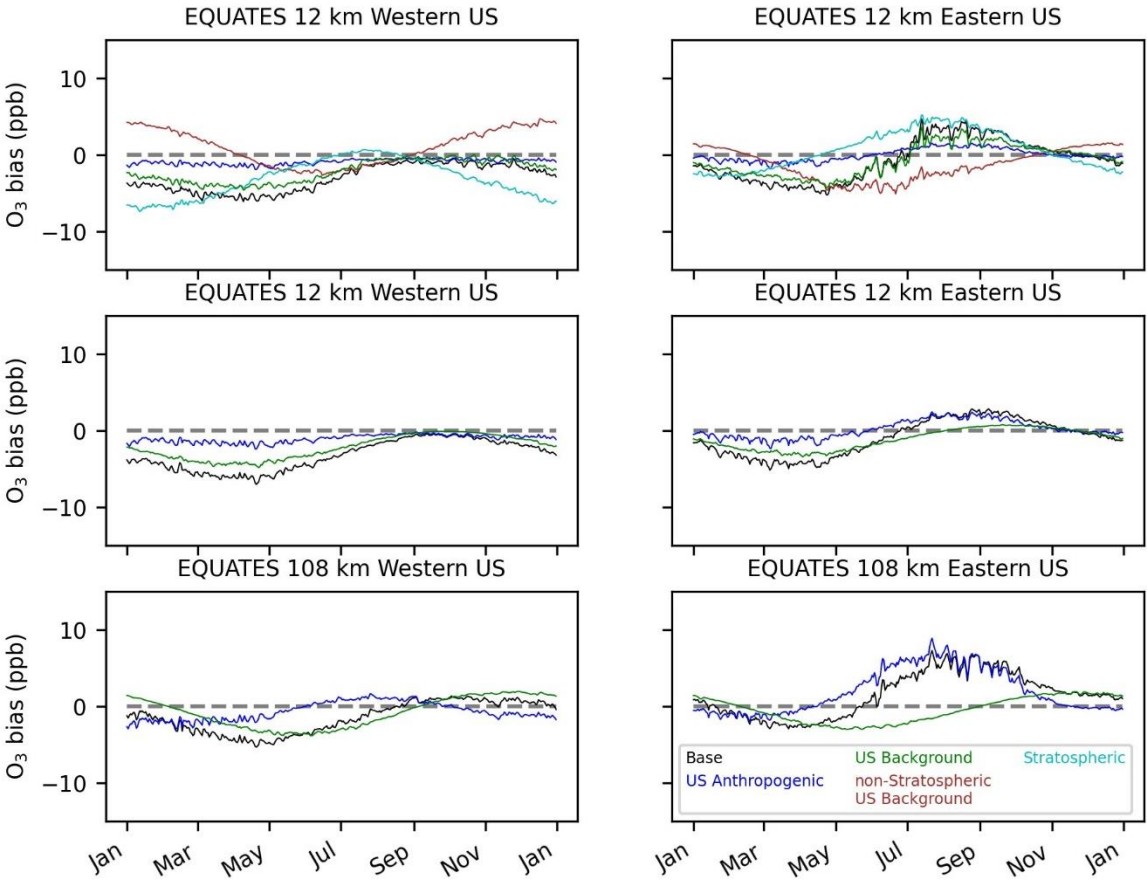

**Figure 11. Daily average of inferred MDA8 O₃ model bias from EQUATES CMAQ simulations averaged across US model grid cells in the eastern and western US. A longitude of 97 °W is used as the dividing line between east and west. EQUATES O₃ concentrations include base O₃ as well as O₃ components from US anthropogenic, non-stratospheric US background, and stratospheric sources for 12 km. For both the 12 km and 108 km simulations, O₃ concentrations of base, US anthropogenic, and total US background are also shown. For the case with multiple US background O₃ components, US background indicates the sum of biases for individual US background components.**

## 3.4 CTM biases by O₃ concentration

The contributions and biases of different O₃ components have been presented so far as annual or seasonal averages (Figures 2-3, 5-6, 8, and 10), as the fourth highest value which is relevant from a

regulatory perspective (Figures 4 and 7), or as daily averages over US model grid cells (Figures 9 and 11). However, the relative contributions of $O_3$ components at different total $O_3$ concentrations is also of interest. For example, the relative contribution of US anthropogenic and US background $O_3$ to total $O_3$ may be different on days with higher total $O_3$ vs. days with lower total $O_3$. Situations where $O_3$ exceeds the NAAQS, which is currently set at a level of 70 ppb, are of particular interest. We analyze the different $O_3$ components at $O_3$ monitoring sites under cases when $O_3$ is less than 60 ppb, between 60 and 70 ppb (inclusive), and greater than 70 ppb. These concentration bins are selected because they reflect the current level of the standard (70 ppb) as well as a potential range which might be considered as the level of the standard in the future (60-70 ppb). We compare the results of the analysis when using both simulated and observed $O_3$ bins. Simulated $O_3$ has a positive bias on average when simulated $O_3$ is high and has a negative bias on average when observed $O_3$ is high, so selection bias influences these results. For this analysis, we consider the 12 km resolution simulations for the PA and EQUATES simulations. The 12 km simulations are the resolution that is typical for simulations that support regulatory analyses. Monitoring sites are split into western or eastern US using a longitude of 97 °W as the dividing line. The division to western and eastern US is done because there are differences in the contribution of US anthropogenic vs. background contributions in the two parts of the country.

The impacts of the linear regression adjustment technique at the observation sites are examined by comparing the original simulated bias to the residual bias (i.e., the sum of the adjusted individual $O_3$ components minus observed $O_3$) (Figure 12). The change in bias from the original to residual bias is the inferred bias that has been referenced elsewhere. In all cases when $O_3$ is binned by simulated $O_3$ levels, the adjustment brings the bias closer to zero. In the eastern US, high biases at higher simulated $O_3$ levels were reduced for both the PA and EQUATES simulations. In the western US, low biases when simulated $O_3$ was below 60 ppb were brought closer to zero for both the PA and EQUATES simulations. At higher simulated $O_3$ levels, the PA simulations originally had high biases in the western US which were reduced in the adjusted results while the EQUATES simulations originally had low biases in the western US which were improved in the adjusted results. The effects on bias when binning by observed $O_3$ are mixed. In both the western and eastern US for both the PA and EQUATES simulations, the simulations were originally biased low at higher observed $O_3$ levels, with the EQUATES simulations more biased low than

the PA simulations. The low bias is improved in the EQUATES simulations, but in the PA simulations the bias is either about the same or becomes more biased low. The inability of the adjustment to improve the bias across the range of both observed and simulated O₃ levels is a limitation of this technique. The

fitting of multi-axis (lat, lon, season) linear correction factors ($\alpha_i$) will be strongly influenced by the larger population of lower (O₃ < 70 ppb) concentrations and will only correct the upper end if the bias structure is consistent.

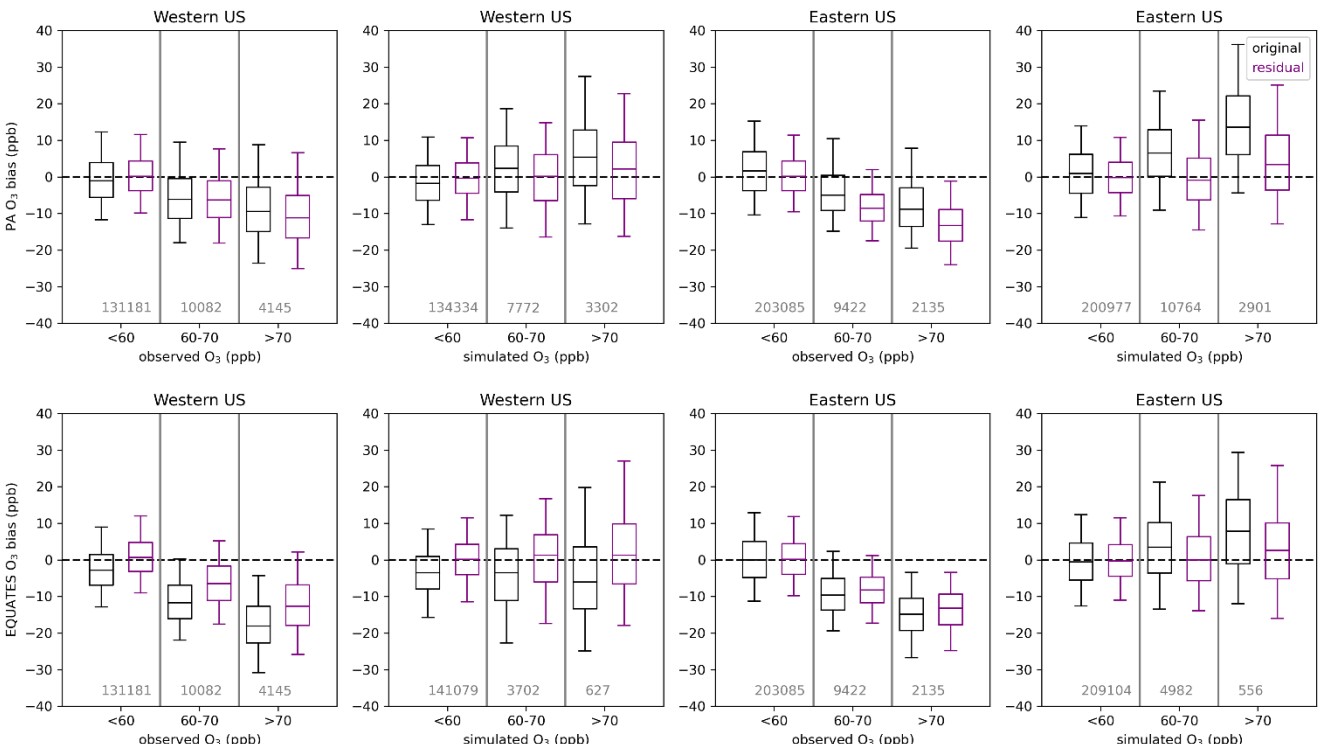

**Figure 12. Bias compared to MDA8 O₃ observations of original simulations (black) and residual bias (purple) obtained as the difference between adjusted MDA8 O₃ and observations for PA (top row) and EQUATES (bottom row) simulations. The horizontal line shows the median; the box shows the 25th-75th percentiles; the whiskers show the 5th and 95th percentiles. Grey vertical lines separate the boxplots for each MDA8 O₃ concentration bin. The numbers at the bottom of each**
**panel are the number of data points falling within each concentration bin.**

For the PA simulations, the contribution from US anthropogenic $O_3$ tends to increase with higher simulated $O_3$ and with higher observed $O_3$ (Figure 13), indicating that domestic anthropogenic pollution is driving the highest $O_3$ concentrations. The contribution from US anthropogenic $O_3$ is higher at eastern US sites than at western US sites due to higher anthropogenic precursor emissions in the east. There may also be impacts on US anthropogenic $O_3$ in the eastern US from $O_3$ or precursor pollutants transported from the western to eastern US. The median US anthropogenic $O_3$ contribution is biased high (+1 ppb in the western US; +4 ppb in the eastern US) when base $O_3$ is between 60 and 70 ppb with higher median biases (+2 ppb in the western US; +6 ppb in the eastern US) when base $O_3$ exceeds 70 ppb. When observed $O_3$ is between 60 and 70 ppb, the median US anthropogenic $O_3$ contribution is biased slightly low in the western US (-0.2 ppb) and biased high in the eastern US (+2 ppb). Bias is higher in the western US when observed $O_3$ exceeds 70 ppb (+1 ppb) but is about the same in the eastern US (+2 ppb). Inferred biases of US anthropogenic $O_3$ are higher across the range of simulated and observed $O_3$ levels in the eastern US compared to the western US.

In the western US, natural $O_3$ tends to be higher when either simulated or observed $O_3$ is greater than 60 ppb; however, the distribution of natural $O_3$ when $O_3$ is above 70 ppb is similar to the distribution of natural $O_3$ when $O_3$ is between 60 and 70 ppb. In the eastern US, the distribution of natural $O_3$ is similar across the range of simulated and observed $O_3$ concentration bins but is slightly higher when $O_3$ is greater than 60 ppb. Long-range international makes a small contribution to $O_3$ across concentration bins and tends to be lower as simulated or observed $O_3$ increases. Canada+Mexico $O_3$ is typically very small and only makes significant contributions at a few near-border sites (not shown). The natural and long-range international $O_3$ components are biased slightly low at monitoring sites in the western US. For western US sites, the sum of the median biases in US anthropogenic and US background (i.e., natural + long-range international + Canada+Mexico) $O_3$ at monitoring sites is negative across the simulated and observed $O_3$ concentration bins but gets closer to zero at higher $O_3$ levels. For eastern US sites, the bias in US anthropogenic $O_3$ is predicted to be the main contributor to biases at high simulated $O_3$ when simulated $O_3$ concentrations exceed 60 ppb. When the $O_3$ components are binned by observed $O_3$ rather than simulated $O_3$, the sum of the median biases in US anthropogenic and US background $O_3$ at monitoring sites in the eastern US is negative across the range of simulated $O_3$ with US background $O_3$

becoming less negatively biased as observed O$_3$ increases and US anthropogenic O$_3$ becoming more positively biased as observed O$_3$ increases.

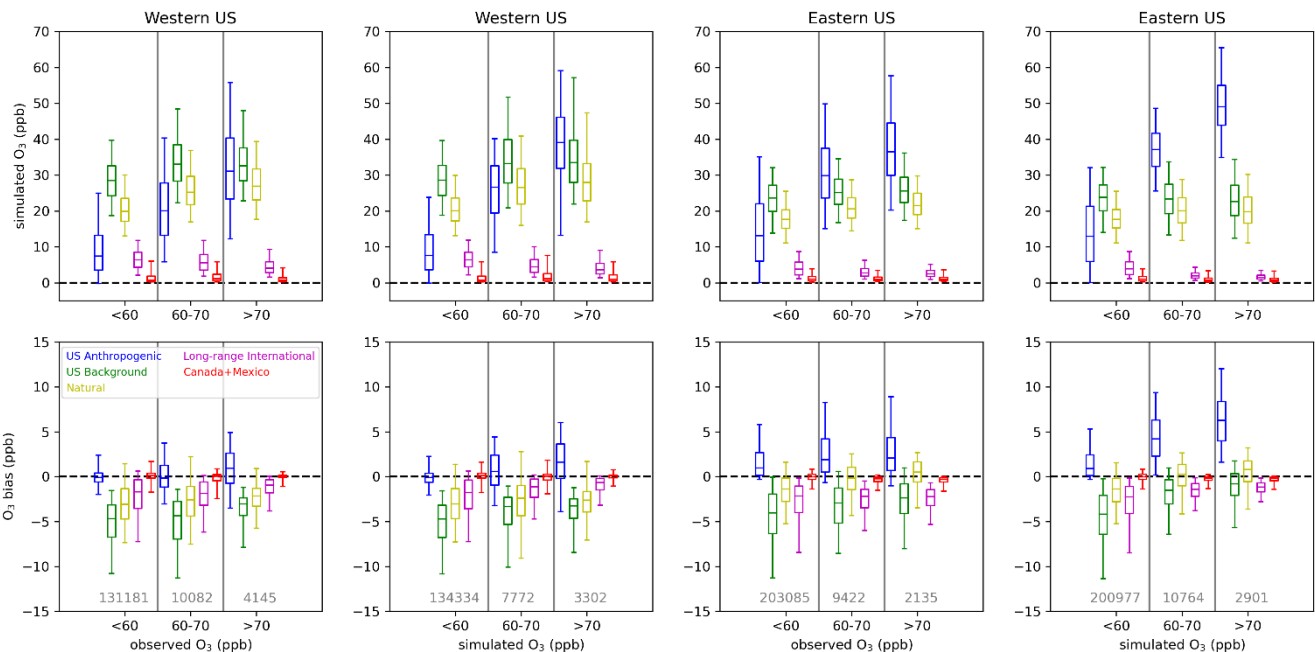

**Figure 13. Contributions to MDA8 O$_3$ from the PA simulation (top row) and inferred biases (bottom row) of US anthropogenic, natural, long-range international, and Canada+Mexico separated by both observed and simulated base MDA8 O$_3$ concentration at O$_3$ monitoring sites. The sum of natural, long-range international, and Canada+Mexico is shown as US background. The horizontal line shows the median; the box shows the 25th-75th percentiles; the whiskers show the 5th and 95th**
**percentiles. Grey vertical lines separate the boxplots for each MDA8 O$_3$ concentration bin. The numbers in the bottom row of panels are the number of data points falling within each concentration bin.**

For the 12 km EQUATES simulations, the US anthropogenic O$_3$ contribution is similar to the 12
770 km PA results across the simulated O$_3$ concentration bins (Figure 14). At higher observed O$_3$, the EQUATES simulations generally simulate lower US anthropogenic O$_3$ compared to the PA simulations. Like in the PA simulations, the US anthropogenic O$_3$ contribution increases with increasing simulated and observed O$_3$, meaning that domestic anthropogenic emissions are mostly driving the highest O$_3$ levels.

There is an inferred negative bias in US anthropogenic $O_3$ in the western US which becomes increasingly more negative as simulated or observed $O_3$ increases. In the eastern US, there is an inferred positive bias in US anthropogenic $O_3$ which becomes larger at higher simulated $O_3$ concentrations (median bias of +0.05, +2, +4 ppb at <60, 60-70, and >70 ppb simulated $O_3$). There is also an inferred high bias across the range of observed $O_3$; however, the magnitude is smaller, and the bias does not increase much at higher levels of observed $O_3$ (median bias of +0.05, +0.5, and +0.6 ppb at <60, 60-70, and >70 ppb observed $O_3$).

The contribution from stratospheric $O_3$ is higher in the western US than in the eastern US across simulated and observed $O_3$ concentrations. In the western US, stratospheric $O_3$ tends to be higher when either observed or simulated is above 60 ppb. In the eastern US, stratospheric $O_3$ is at similar levels across the range of simulated and observed $O_3$. In the western US, stratospheric $O_3$ has a negative bias which gets closer to zero when simulated and observed $O_3$ is above 60 ppb. In the eastern US, stratospheric $O_3$ has a positive bias which gets higher when simulated and observed $O_3$ are above 60 ppb. In both the western and eastern US, non-stratospheric US background $O_3$ makes similar contributions across different $O_3$ concentrations. In the western US, non-stratospheric US background $O_3$ has a negative bias when simulated or observed $O_3$ is below 60 ppb and a positive bias when $O_3$ is above 60 ppb. In the eastern US, non-stratospheric US background $O_3$ has a negative bias across the range of simulated and observed $O_3$. The magnitude of the negative bias is smaller when simulated or observed $O_3$ is below 60 ppb than when $O_3$ is above 60 ppb.

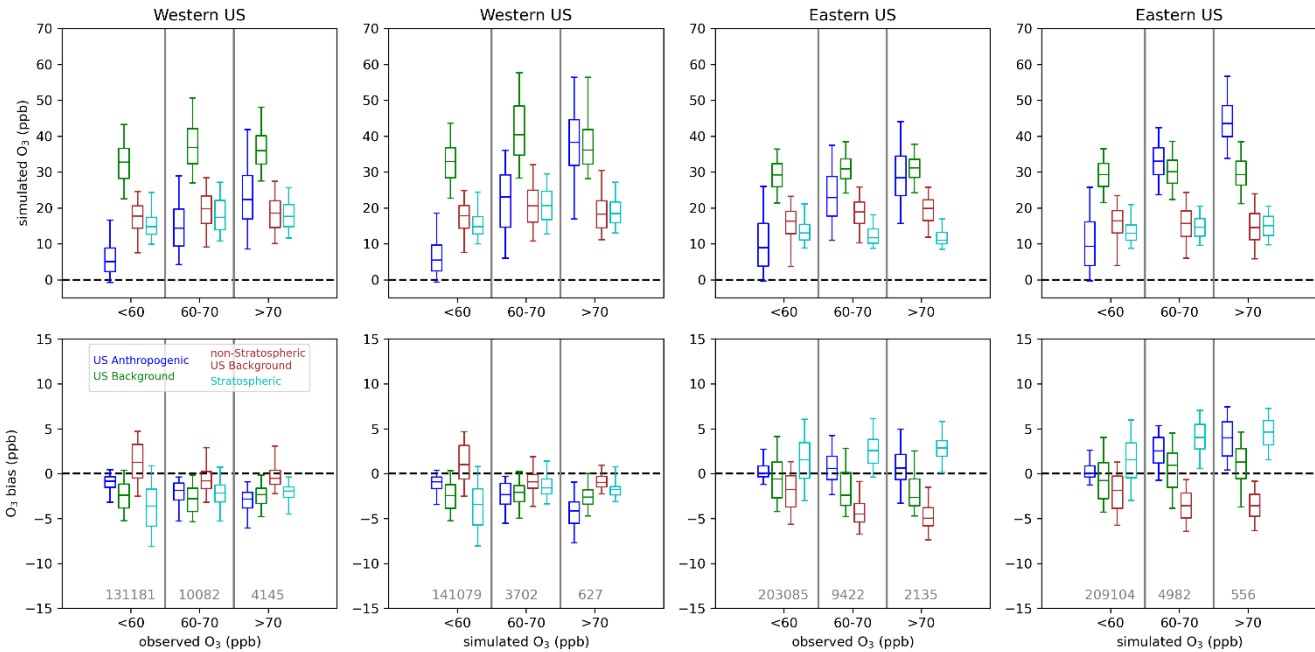

**Figure 14. Contributions to MDA8 O₃ by the EQUATES simulation (top row) and inferred biases (bottom row) of US anthropogenic, US background no stratospheric, and stratospheric separated by both observed and simulated base MDA8 O₃ concentration at O₃ monitoring sites. The sum of non-stratospheric US background and stratospheric is shown as US background. The line shows**
**the median; the box shows the 25th-75th percentiles; the whiskers show the 5th and 95th percentiles. Grey vertical lines separate the boxplots for each MDA8 O₃ concentration bin. The numbers in the bottom row of panels are the number of data points falling within each concentration bin.**

Binning the O₃ contributions and inferred biases by observed and simulated O₃ results in different numbers of data points in each sample. In the western US, there were 4145 instances when observed O₃ exceeded 70 ppb, while there were 3302 (PA) and 627 (EQUATES) instances when simulated O₃ exceeded 70 ppb at a monitoring site, with a large fraction of the observed and simulated exceedances occurring in California. In the eastern US there were 2135 instances when observed O₃ exceeded 70 ppb

with 2901 (PA) and 556 (EQUATES) instances when simulated O₃ exceeded 70 ppb. The PA simulations more accurately simulated the number of exceedances compared to EQUATES, though this does not consider the timing or location of exceedances. Given the different number of samples in the observed

vs. simulated bins and the lower number of data points for EQUATES simulated $O_3$ exceeding 70 ppb, it is possible that the population of data points are when simulated $O_3$ exceeds 70 ppb are not spatially representative of the population when observed $O_3$ exceeds 70 ppb.

For the western US, the PA simulations largely capture the spatial distribution of exceedances seen in the observations, although the number of exceedances is underestimated (Figure 15). The exceedances from the EQUATES simulations are not very representative of the spatial distribution of observed exceedances in the western US as there are very few sites with more than one or two exceedances outside of California. In particular, the number of exceedances in the Denver, Colorado; Phoenix, Arizona; Las Vegas, Nevada; and Boise, Idaho; areas are underestimated in EQUATES relative to both the PA simulations and observations. Both the PA and EQUATES simulations underestimate the number of exceedances in the state of Utah. For the eastern US, the PA simulations generally capture the spatial distribution of observed exceedances but simulate too many exceedances. This is particularly notable in the northeastern US and along the Gulf Coast. The EQUATES simulations underestimate the number of exceedances, although the spatial distribution is generally similar to the observations. The degree of spatial representativeness provides additional context for interpreting the findings for the $O_3$ component contributions and biases binned by $O_3$ levels. For the western US, the findings for instances when $O_3$ exceeds 70 ppb are not applicable to the western US more broadly. There are a limited number of instances when $O_3$ exceeds 70 ppb in the western US outside of California. These results are mostly indicative of conditions in the Los Angeles area and in the Central Valley in California. This applies especially to the EQUATES results, but it is also the case for the PA simulations and the observations. For the eastern US, on the other hand, there is enough spatial variability in the observations as well as both sets of simulations to interpret the findings for the eastern US more generally. These results are informative in an average sense but are not expected to hold in all cases when applied to specific monitoring sites or to specific days (e.g., fourth highest $O_3$). The biases for bins 60-70 ppb and greater than 70 ppb should be interpreted with caution because the inferred biases apply the mean tendency to these high concentration subpopulations.

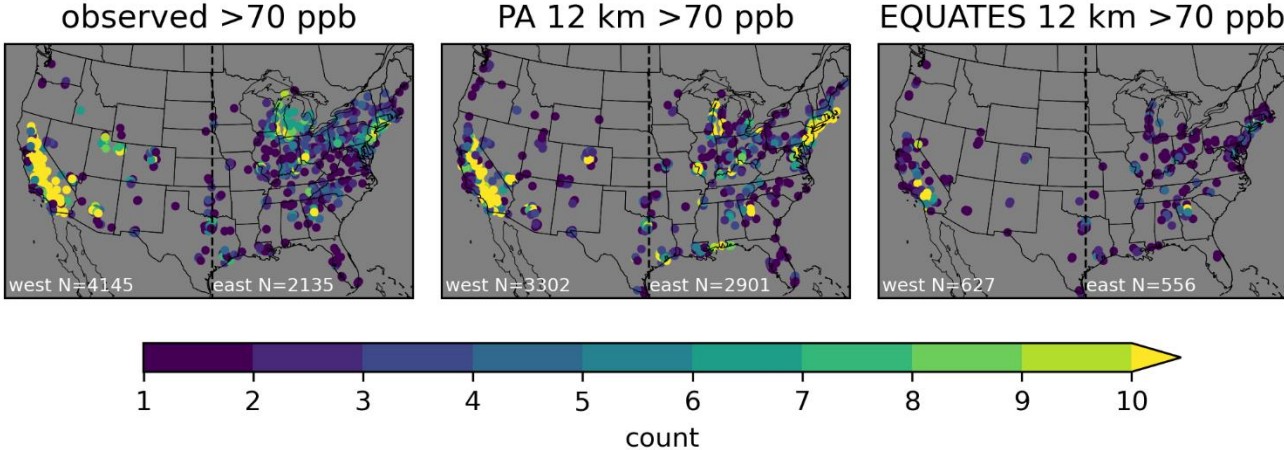

**Figure 15. Spatial distribution of the number of times MDA8 O₃ exceeded 70 ppb for observed and simulated O₃. The circles show the locations of sites, and the color indicates the number of times MDA8 O₃ exceeds 70 ppb at each site for observations (left), PA 12 km simulation (middle), and EQUATES 12 km simulation (right). Only sites with at least one exceedance are shown. The black dotted line shows the longitude of 97° W which is used to divide west and east. Similar results for other model resolutions are shown in Figure S13.**

## 4 Conclusions

In this work, we use two sets of CMAQ simulations to analyze the contributions to US background O₃ from different sources. Naturally occurring sources, long-range international anthropogenic pollution, and short-range international anthropogenic pollution from Canada and Mexico are separately considered for one set of simulations. In the other set of simulations, stratospheric and non-stratospheric sources of US background O₃ are also considered separately. We also consider the contribution to total O₃ from US domestic anthropogenic sources. The measurement-model data fusion approach for apportioning bias to US anthropogenic and US background O₃ components from our previous study (Skipper et al., 2021) was extended to identify biases in separate US background O₃ components. The results generally confirm previous high-level results, but provide new insights from additional components and more detailed analysis.

Results indicated that US anthropogenic $O_3$ was consistently inferred to be biased high (on an annual and seasonal average basis) in the eastern US where domestic anthropogenic emissions are the dominant contributor to total $O_3$, with increasingly higher biases with coarser model resolution and at higher simulated $O_3$ concentrations. This is consistent with our previous findings. This does not necessarily imply that the trend of decreasing biases with finer resolutions would continue at resolutions finer than 12 km as we have not tested this approach at those resolutions. As noted in Section 3.3, previous modeling studies examining the effects of horizontal resolution have found that $O_3$ increased over urban areas with finer resolution, so the findings for the effects of model resolution should be taken to apply our current results rather than as a general finding on the impacts of model resolution. Our finding that US anthropogenic $O_3$ biases increase with higher $O_3$ does not hold when $O_3$ is binned by observed rather than simulated concentrations. There is much less variation in the US anthropogenic $O_3$ bias across the range of observed $O_3$ than for simulated $O_3$. Although the choice of binning $O_3$ by observed or simulated levels changes the sample of data, the results for the eastern US are generalizable to this part of the country because the samples have consistent spatial representation across the eastern US. In the western US, US anthropogenic $O_3$ was inferred to be biased high at higher $O_3$ levels for the PA simulations and biased low at higher $O_3$ levels for the EQUATES simulations. These differences are explained by the use of different emission inventories in the two sets of simulations. Regardless, the findings for inferred $O_3$ biases at higher $O_3$ levels in the western US are not broadly applicable to the entire western US because the sample that these findings are based on is dominated by sites in California. There are relatively few sites in other states in the western US that contribute to this sample, so the results are not likely to be indicative of conditions in other parts of the western US.

The correction of US background components provided consistent results with previous studies, but more detail. Like Skipper et al. (2021) and Hosseinpour et al. (2024), simulated US background $O_3$ was inferred to be biased slightly low overall. The original simulated annual averages of US background $O_3$ across all the PA and EQUATES modeling configurations considered here ranged from 30-33 ppb while the adjusted annual average US background $O_3$ ranged from 31-34 ppb. The annual average of simulated US background $O_3$ for the hemispheric-scale (108 km resolution) and continental-scale (12 km resolution) modeling was slightly higher for the EQUATES simulations (32-33 ppb) than for the PA

simulations (30-31 ppb). The differences are not explainable by the updated chemical mechanism used in EQUATES because the most relevant updates (halogen-mediated $O_3$ loss) tend to reduce $O_3$ at the northern mid-latitudes (Sarwar et al., 2019; Appel et al., 2021). The difference is also not likely due to anthropogenic emissions outside of the US, which are similar between the two sets of simulations. So, the higher US background $O_3$ in EQUATES likely relates to differences in the natural emissions. The EQUATES simulations used MEGAN for biogenic emissions throughout the entire Northern Hemisphere while the PA simulations used BEIS for biogenic emissions in North America and MEGAN elsewhere. The two hemispheric model configurations also used different sources for soil $NO_x$ emissions (see Section 2.1) which could contribute to differences in US background $O_3$. Lightning $NO_x$ emissions were the same in EQUATES and PA hemispheric-scale simulations, but the continental-scale PA simulations did not include lightning in the continental domain. Given that US background $O_3$ in both the EQUATES and PA 12 km continental-scale simulations are 1 ppb lower than their northern hemispheric counterparts, the differences in US background $O_3$ in the continental-scale simulations is more likely driven by the large-scale background inherited through the lateral boundary conditions than from differences in lightning $NO_x$ configurations.

This work separated US background $O_3$ into natural, short-range international, and long-range international components, and each had distinct seasonality to the inferred bias. Short-range international (Canada+Mexico) was marginally high-biased in spring/winter and marginally low-biased in summer. The contribution from natural and long-range international $O_3$ have larger seasonality, which are slightly out of phase. Natural $O_3$ bias was low in winter, but high in summer peaking in July. Long-range international $O_3$ was consistently low-biased with a minimum in April and a maximum (near unbiased) in August-September. From May to October, the natural and long-range international $O_3$ biases are largely offsetting while they are reinforcing in other parts of the year.

The seasonality of inferred long-range international bias highlights a key uncertainty in correlative bias attribution. The biases associated with long-range international may be misattributed due to the difficulty of the regression model formulation to isolate stratospheric influences from other natural sources such as lightning and soil $NO_x$, wildfires, and biogenic VOC emissions, all of which have a high degree of uncertainty. Stratospheric $O_3$ is expected to have similar temporal and spatial patterns to long-

range international $O_3$, with contributions being higher in spring and at high elevations. It is suspected that the regression model formulation may be assigning a negative bias in long-range international $O_3$ to make up for missing stratospheric $O_3$ that has a similar pattern to long-range international $O_3$ while at the same time assigning a high bias for natural $O_3$ to reallocate some of stratospheric $O_3$ that is present in natural $O_3$ to long-range international $O_3$ instead. Results for the stratospheric $O_3$ tracer in the second set of simulations support the idea that there is missing stratospheric $O_3$ at the surface level in the western US as the stratospheric $O_3$ is inferred to be biased low. Taken together, there is an overall low bias in the simulated US background $O_3$ that is most pronounced in the spring. This may be a result of too little stratospheric $O_3$ reaching the surface. Photolysis of particulate nitrate over oceans has been found to increase $O_3$ (Shah et al., 2023; Sarwar et al., 2024). This process is not included in the chemical mechanism which could contribute to low biases in $O_3$ during the same time of year. The potential for misattribution is not specific to the methods employed here but is inherent to correlative bias approaches with incomplete information contained in independent variables.

Analysis of the original bias and residual bias emphasize the importance of subpopulation diversity. The correction factors are optimized for the whole population and can degrade performance at any subpopulation (e.g., a site, a day, or a subgroup). For example, in the western US, the PA simulation was originally high-biased for days with high predictions and low-biased for days with high observations (>70 ppb). The overall correction was downward for both populations because they are generally consistent spatially and seasonally. This means that the "corrected" model has more bias on days with high observations in the western US than the "uncorrected." This is not unexpected but highlights that correlative adjustments should be considered as broad conclusions and should only be cautiously applied more narrowly (e.g., specific monitors or days). This is a limitation of the linear formulation as noted by Hosseinpour et al. (2024).

This work has focused only on surface $O_3$. We are not able to draw a conclusion as to whether the potential lack of stratospheric $O_3$ is a result of biases in the UTLS PV scaling in the upper layers or from errors in vertical transport from upper layers to the surface. More detailed studies that analyze the entire vertical structure, such as a recent study of CMAQ stratospheric $O_3$ by Itahashi et al. (2020), are needed to identify the exact causes and solutions for the surface biases identified here. Another potential area for

future work is to separate stratospheric $O_3$ from natural sources in a set of simulations like those conducted for the $O_3$ Policy Assessment. This might solve the suspected issue of bias in stratospheric $O_3$ being allocated to long-range international emissions that may be caused by the correlation of stratospheric $O_3$ and long-range international impacts. While details on the spatial and temporal characteristics of biases in different $O_3$ components are provided here, the correlational bias attribution method employed here does not necessarily identify the specific factors that drive the biases. These results provide estimates of potential biases in US background and US anthropogenic $O_3$ that can inform more targeted future work examining the individual sources in greater detail. Additional future work could take a process-oriented approach rather than the source-oriented approach described here. A process-oriented approach would focus on how different physical and chemical processes (deposition, transport, photochemical activity, etc.) relate to biases in $O_3$ simulations. The role of uncertainties in $O_3$ deposition and in $O_3$ production efficiency across various chemical regimes could be examined in a more process-focused analysis. A further area for future work is to apply the data fusion bias correction method to an ensemble of US background $O_3$ estimates from different models. This work has only used the CMAQ model. A test of the method would be to apply it to several different models to determine whether it is able to reduce the uncertainty of US background $O_3$ estimates while also reducing bias in total $O_3$.

**Acknowledgements**

TNS and AGR received funding from the Phillips 66 Company. AGR also received funding from NASA HAQAST. Funding organizations have not dictated the topic or content of this work nor have they had any editorial role. The views expressed in this paper are those of the authors and do not necessarily represent the view or policies of the U.S. Environmental Protection Agency, the Phillips 66 Company, or NASA. We thank Benjamin Murphy and Sergey Napelenok for their comments on a draft version of the paper.

## Code and data availability

The CMAQ source code is available from GitHub (https://github.com/USEPA/CMAQ) and Zenodo (https://zenodo.org/doi/10.5281/zenodo.1079878). $O_3$ observational data are available via the AQS website (https://www.epa.gov/aqs).

## Author contributions

TNS: conceptualization, investigation, methodology, software, visualization, writing – original draft. CH: data curation, software, writing – review and editing. BHH: data curation, software, writing – review and editing. RM: software, writing – review and editing. KMF: data curation, software, writing – review and editing. AGR: conceptualization, methodology, resources, supervision, writing – review and editing.

## Competing interests

The authors declare that they have no competing interests.

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
