# Peer review of "Source specific bias correction of US background and anthropogenic ozone modeled in CMAQ"

_EGUsphere, 2024_

## Author Comment (AC1)

We would like to thank the reviewers for their comments. Below are responses to the individual comments. Responses are in blue, and revised text in the manuscript is in red.

**Reviewer 1:**

Background  $O_3$  constitutes a significant portion of total surface  $O_3$  and the contribution becomes higher when  $O_3$  levels decline. This study used a measurement-model data fusion approach to assess CTM biases in USB  $O_3$  and attribute these biases to different sources. Two sets of CMAQ simulations, PA and EQUATES, are conducted to analyze the contributions of different sources to USB  $O_3$ . While the study is of scientific significance in assessing model biases in background  $O_3$  estimation, two major concerns need to be addressed by the authors.

(1) The two sets of simulations are for different years, the model configuration and inputs are different, different versions of emissions inventory are adopted, which will introduce substantial uncertainties. The PA simulations cover the year of 2016; while the EQUATES simulations span between 2002-2019. Additional simulations using the EQUATES modeling framework were conducted for 2016–2017 to estimate USB O3 and USA O3 using the zero-out method. CMAQ v5.2.1 was used for the PA simulations while CMAQ v5.3.2 was used for the EQUATES simulations. It seems this study is not outlined within a comprehensive framework but combine different modelling work to do the current study. The differences of biases caused by different model configurations and model setup between the two sets of scenarios need to be fully discussed. (2) While biases in the study are fully discussed, the reasons behind these biases remain ambiguous. Factors such as uncertainties in emissions inventory, meteorology simulations, and chemical mechanisms could contribute to biases. Providing insights into the main drivers of biases and offering suggestions for modelers to mitigate these biases would enhance the value of the study for readers seeking to improve background O3 modeling accuracy.

(1) It is true that the Policy Assessment (PA) and EQUATES simulations were not conducted specifically for this study. However, an opportunity to use these datasets for the analysis presented here was available. Some major differences between the two sets of simulations that are expected to affect  $O_3$  are given in the final paragraph of the original introduction. One is the addition of halogen chemistry which was added to CMAQ between the model version used for the PA simulations and the model version used for the EQUATES simulations. The second is a different US anthropogenic emissions inventory. Further details on model configuration of both sets of simulations are given in Tables S4-S5. See the response to comment below (2. Methods) for additional details that have been added to describe the model configuration.

(2) Agreed that it would be great if we were able to identify the specific reasons for biases here, but we are not able to assess that with confidence in the current study. This work is intended to be largely descriptive rather than proscriptive. The following additional discussion has been added to Section 4 to clarify and expand on this:

"While details on the spatial and temporal characteristics of biases in different  $O_3$  components are provided here, the correlational bias attribution method employed here does not necessarily identify the specific factors that drive the biases. These results provide estimates of potential biases in USB and USA  $O_3$  that can inform more targeted future work examining the individual sources in greater detail."

**Specific comments:**

1. **Abstract**. The current form of the abstract is notably objective, it lacks quantitative results detailing the biases and their spatial-temporal characteristics. Additionally, what can we do to reduce these biases? The abstract could benefit from discussing potential strategies to mitigate these biases.

We have added the following quantitative results to the abstract:

"Summer average US anthropogenic  $O_3$  in the eastern US was estimated to be biased high by 2, 7, and 11 ppb (11%, 32%, and 49%) for one set of simulations at 12, 36, and 108 km resolutions and 1 and 6 ppb (10% and 37%) for another set of simulations at 12 and 108 km resolutions."

**and**

"Despite this, results indicate a negative bias in modeled estimates of the impact of stratospheric  $O_3$  at the surface, with a western US spring average bias of -3.5 ppb (-25%) estimated based on a stratospheric  $O_3$  tracer."

See item (2) of the previous comment for a response on the discussion of potential strategies to mitigate the biases.

**2. **Methods**. The overall model configuration needs to be briefly outlined in the main text, such as modelling domain, WRF configuration, CMAQ gas-phase mechanism, IC/BC, vertical layers, etc.**

Many of these details are available from Tables S4 and S5. Information on the vertical layer structure and more details on the modeling domains have now been added to Table S5. We have also added additional details given below in Section 2.1 of the manuscript. (Note this response is identical to the response to reviewer #2 comment marked "Table 1")

Vertical structure:

[revised manuscript text omitted]

We have also slightly updated the description in Table 1 of the ZANTH simulation to clarify that this represents zeroing out anthropogenic emissions globally and have added references to the simulation names given in Table 1 in Section 2.1 to further connect the text of this section to Table 1. With these clarifications, the suggestion of a new table with check and x markings should not be necessary.

It is intentional that Table 1 and S1 are the same. The reason is explained in the caption to Table S1 which is repeated below (relevant sentence in bold here for emphasis):

"Table S1. Simulation names and descriptions for hemispheric-scale and regional-scale simulations. Table adapted from 2020  $O_3$  Policy Assessment Table 2-1 (USEPA, 2020). Table S1 is reproduced from Table 1 in the main text to aid in interpreting Tables S2 and S3."

**1. In the main text, Tables S4 and S5 (Line 95) comes before Table S3 (Line 101), this is strange.**

We have slightly rearranged the text to make the first mentions of Tables S3, S4, and S5 appear in order.

**2. Line 115, "STRAT" should be spelled out at the first time it appears.**

We have made a small change to the relevant sentence to define the meaning of STRAT.

Original: "We refer to the PV tracer concentrations as STRAT  $O_3$  since it relates to the stratospheric influence, but it only partly replicates the impact of stratospheric O3 since it does not undergo chemical losses."

Revised: "We refer to the PV tracer concentrations as STRAT (short for stratospheric) O3 since it relates to the stratospheric influence, but it only partly replicates the impact of stratospheric O3 since it does not undergo chemical losses."

**3. **CTM results**: can you explain further why the 12km simulations have the best performance? Additionally, what about the model performance for NO2 simulations?**

The original statement in the manuscript that the 12 km simulations have the best performance as indicated by the normalized mean bias is incorrect, as the 108 km EQUATES simulations have normalized mean bias (NMB) closer to zero compared to the 12 km EQUATES simulations. This sentence has therefore been removed, and we now simply present the performance for each simulation as summarized by the NMB. The role of different  $O_3$  components in these biases is further discussed in Sections 3.3 and 3.4.

While the performance for  $NO_x$  is of course relevant for  $O_3$ , a full performance evaluation for  $NO_2$  or  $NO_x$  is considered beyond the scope of the current paper.

**4. Table 2 and 3, what's the unit of the data in this table?**

The following has been added to the captions of Tables 2 and 3:

"Numbers in the table are in units of ppb."

5. **section 3.3.** The authors extensively compared the differences in deviations between different scenarios. Are these differences in deviations related to the zero-out method neglecting nonlinear ozone production?

It is not clear that the nonlinear ozone production which is not accounted for in the zero-out method would result in differences in the inferred biases between the PA and EQUATES modeling setups. Both setups use the zero-out method to estimate US background and US anthropogenic contributions. We have added some additional details in Section 2.1 to describe how the zero-out method neglects non-linear interactions between different emissions sources and why it is preferred to other techniques here:

"The zero-out method is the most common approach for simulating USB  $O_3$ , though other approaches such as sensitivity simulations and source tagging techniques have also been previously employed (Jaffe et al., 2018). The zero-out method neglects non-linear interactions between sources which can affect the simulated source contribution (Wu et al., 2009; Dolwick et al., 2015). However, the zero-out method is consistent with the definition of USB  $O_3$  as the level of  $O_3$  in the absence of US anthropogenic emissions, while sensitivity or tagging techniques would instead provide an estimate of source contributions to total simulated O3 (including O3 from US anthropogenic sources)."

6. **section 3.4.** The authors extensively described the results of model bias. These results seem to be an extension of section 3.3, but what can these results further illustrate?

The following additional motivation for Section 3.4 has been added to the beginning of Section 3.4:

"The contributions and biases of different  $O_3$  components have been presented so far as annual or seasonal averages (Figures 2-6 and 8) or as daily averages over US model grid cells (Figures 7 and 9). However, the relative contributions of  $O_3$  components at different total  $O_3$  concentrations is also of interest. For example, the relative contribution of USA and USB  $O_3$  to total  $O_3$  may be different on days with higher total  $O_3$  vs. days with lower total  $O_3$ ."

Additionally, this section focuses on the impacts at ozone monitoring sites, which are relevant for regulatory purposes. This is already stated in Section 3.4 so no additional text is needed to describe this.

**Reviewer 2:**

**Major comments:**

1. This study analyzed surface ozone from a suite of hemispheric and regional-scale CMAQ simulations for 2016 and 2017 and attempted to attribute the biases in model simulated total surface ozone to different components, including ozone produced from US anthropogenic emissions, natural sources, intercontinental transport, and stratospheric intrusions. Understanding US background ozone and its components is of broad interest because they are directly relevant to the setting and implementation of US ozone air quality standards. However, the manuscript needs to be substantially revised before it can be published. Description of the methodology used and discussions in many sections are incomplete. The authors should also discuss the model biases in the context of published literature. The referee's main concern is on the methodology used to attribute the model biases to different components. The description of the data fusion model in Section 2.3 is hard to understand. Is the data fusion model trained using one set of simulations and applied to another set of simulations for the bias attribution? How do you know the sources of biases in the two sets of simulations are the same? There are a couple of places where the authors refer to Skipper et al. (ES & T, 2021) for the method, but that study did not discuss the different USB components.

The description of the data fusion model approach in Section 2.3 has been reorganized and expanded as shown below to clarify:

Original: "Each  $O_3$  component is multiplied by the alpha adjustment factor which varies as a function of space and time. The longitude and latitude terms are intended to capture the spatial variability of  $O_3$  biases while the z term is intended to capture biases in  $O_3$  related to elevation. The sinusoidal day of year terms are intended to capture the cyclical nature of  $O_3$  production and to identify any seasonal dependence in  $O_3$  biases."

Revised: "Each simulated O3 component ( $O_{3i}^{simulated}$ ) is multiplied by the alpha adjustment factor for that component ( $\alpha_i$ ), which varies as a function of space and time, to calculate an adjusted estimate of each O3 component. The inferred model bias for a particular component is calculated as the difference between the original simulated O3 and adjusted O3 for that component. The individual adjusted O3 components are summed to calculate the total adjusted O3. The longitude and latitude terms of  $\alpha_i$  are intended to capture the spatial variability of O3 biases while the z term of  $\alpha_i$  is intended to capture biases in O3 related to elevation. The sinusoidal day of year terms of  $\alpha_i$  are intended to capture the cyclical nature of O3 production and to identify any seasonal dependence in O3 biases."

Regarding the questions "Is the data fusion model trained using one set of simulations and applied to another set of simulations for the bias attribution? How do you know the sources of biases in the two sets of simulations are the same?":

A separate data fusion model is developed for each individual model configuration. Each model is only applied to the particular model configuration that it was trained on. We have updated part of Section 2.3 to clarify this:

Original: "A separate regression model is developed for each model resolution and USB  $O_3$  component split. There are three model resolutions and three USB  $O_3$  splits for the PA simulations, resulting in nine

PA models. There are two model resolutions for the EQUATES simulations. The 12 km EQUATES data has two USB  $O_3$  splits while the 108 km EQUATES data has one USB  $O_3$  split, resulting in three EQUATES models. For the PA models, only 2016 data is used since these simulations are for only that year. The models are trained on both 2016 and 2017 data for the EQUATES data."

Revised: "A separate regression model is developed for each separate model configuration (i.e., model resolution, PA or EQUATES simulation, and USB O3 component split). There are three model resolutions and three USB O3 splits for the PA simulations, resulting in nine PA models. There are two model resolutions for the EQUATES simulations. The 12 km EQUATES data has two USB O3 splits while the 108 km EQUATES data has one USB O3 split, resulting in three EQUATES models. For the PA models, only 2016 PA simulation data are used to train the models since these simulations are for only that year. For the EQUATES models, both 2016 and 2017 EQUATES simulation data are used to train the models."

The suggestion to add more discussion of published literature is addressed in subsequent responses.

2. The title of this paper is about the bias of US background ozone, but in the abstract and in the paper, there is substantial discussion on the biases of US anthropogenic O3 and the influence of model resolution. The authors stated "The estimated correction factors suggest a seasonally consistent positive bias in US anthropogenic O3 in the eastern US, with the bias becoming higher with coarser model resolution and with higher simulated total O3 though the bias does not increase much with higher observed O3." This statement seems to imply that coarser model resolution always produces higher US anthropogenic O3, which is not true. There is clearly a seasonal dependence. During winter when ozone production is in NOx-saturated regime, coarser model resolution leads to artificial dilution of NOx and thus higher O3 due to less NOx titration. During summer, however, when ozone production at most locations is in NOx-limited regime, coarser model resolution may lead to lower ozone concentrations produced from regional anthropogenic emissions. Increasing model resolution may lead to higher simulated US anthropogenic O3, leading to better agreement with observations, such as in the Central Valley of California. These seasonal characteristics of model resolution impacts on US anthropogenic ozone are clearly demonstrated in the published literature, including the recent studies of Schwantes et al. (2021) and Lin et a. (2024).

Schwantes, R. H., Lacey, F. G., Tilmes, S., Emmons, L. K., Lauritzen, P. H., Walters, S., et al. (2022). Evaluating the impact of chemical complexity and horizontal resolution on tropospheric ozone over the conterminous US with a global variable resolution chemistry model. *Journal of Advances in Modeling Earth Systems*, 14(6), e2021MS002889. https://doi.org/10.1029/2021MS002889

Lin, M., L. W. Horowitz, M. Zhao, L. Harris, P. Ginoux, J. P. Dunne, S. Malyshev, E. Shevliakova, H. Ahsan, S. Garner, F. Paulot, A. Pouyaei, S. J. Smith, Y. Xie, N. Zadeh, L. Zhou. *The GFDL Variable-Resolution Global Chemistry-Climate Model for Research at the Nexus of US Climate and Air Quality Extremes.* Journal of Advances in Modeling Earth Systems, in press, https://doi.org/10.1029/2023MS003984, 2024

The title has been updated to "Source specific bias correction of US background and anthropogenic ozone modeled in CMAQ" to reflect that there are findings relevant to both US background and anthropogenic  $O_3$ .

The finding of higher US anthropogenic  $O_3$  biases at higher model resolutions applies to the eastern US, so the statement is not in conflict with the previous findings for  $O_3$  simulated in the Central Valley of California. This statement is not meant to assert that coarser resolution leads to higher biases in all scenarios. It is only meant to apply to the specific findings here for the CMAQ simulations in the eastern US during warmer months. We have added some additional text to the abstract which clarifies the applicability of this finding:

"Summer average US anthropogenic  $O_3$  in the eastern US was estimated to be biased high by 2, 7, and 11 ppb (11%, 32%, and 49%) for one set of simulations at 12, 36, and 108 km resolutions and 1 and 6 ppb (10% and 37%) for a second set of simulations at 12 and 108 km resolutions."

In Section 3.3, where similar results are discussed, we have added the following which includes discussion of findings in the recent articles mentioned by the reviewer:

"The inferred high biases in USA  $O_3$  in the eastern US are primarily driven by biases in the summer and fall (Table S15, Figures S5-S7). Inferred eastern US USA  $O_3$  biases average 2, 7, and 11 ppb in the summer and 3, 4, and 5 ppb in the fall for the 12, 36, and 108 km simulations. In the western US, where USA  $O_3$  is mostly found to be biased low, coarser model resolution results in the summer average bias changing from slightly negative in the 12 km simulations (-0.5 ppb) to slightly positive in the 36 and 108 km simulations (+0.7 ppb and +1.0 ppb).

In contrast to our results showing an increase in O3 with coarser resolution, Schwantes et al. (2022) found that O3 tended to increase for a finer resolution simulation (~14 km vs. ~111 km over the CONUS) during the summer over urban areas using the Community Earth System Model (CESM)/Community Atmosphere Model with full chemistry (CAM-chem) model which was attributed to improvements in the spatial resolution of NOx emissions resulting in less artificial dilution of NOx and enhanced O3 production. Similarly, Lin et al. (2024) found that a variable resolution global model (AM4VR with horizontal resolution of 13 km over the CONUS) had increased O3 over urban areas compared to a fixed resolution model (AM4.1 with horizontal resolution of ~100 km globally). In particular for the Los Angeles Basin and Central Valley regions of California, Lin et al. (2024) found that the increased resolution of AM4VR led to better simulation of observed O3 levels in these areas due the finer resolution model's ability to represent sharp spatial gradients in areas with NOx-limited vs. NOx-saturated O3 production regimes. Given these previous results finding increased O3 with finer resolution simulations, our results here finding higher biases in USA O3 in the eastern US with coarser resolution should be taken to apply specifically to the CMAQ model results described here rather than as a general finding on the impact of model resolution on O3 production."

**Figure 6: the authors should present results for different seasons, not annual averages.**

Our intention is that Figures 6 and 8 convey information about the spatial variability of the inferred model biases while Figures 7 and 9 convey information about the seasonal variability. We have, however, added new figures to the SI showing spatial maps of the seasonal average inferred model biases (Figures S5-S10 in the revised SI) so that these details are available.

Figure 13: What is the horizontal resolution of PA and EQUATES simulations presented in this figure? Are the differences driven by differences in model configurations or model resolution? The authors should show the comparison from the same configuration but at different resolutions.

The simulations referred to in Figure 13 are at 12 km resolution. This has been added to the figure and caption. The purpose of Figure 13 is to add context to the rest of Section 3.4 which deals exclusively with results from the 12 km simulations. Figure 13 is intended to provide additional context about how representative the results shown in Figures 11 and 12 for  $O_3 > 70$  ppb are for the western US and eastern US regions more broadly. While similar results for other model resolutions are of general interest, this would not fit in with the overall theme of Section 3.4 which otherwise deals only with 12 km results. For these reasons, we have added the suggested figure to the SI (Figure S11 in the revised SI) to show the same results as in Figure 13 for 36 km model resolution (PA simulation only) and 108 km model resolution (both PA and EQUATES simulations).

3. Discussion on stratospheric contribution and CMAQ low-O3 bias in spring should be placed in the broader published literature, including those using dynamic stratospheric ozone tracers with explicit stratospheric chemistry and evaluation with intensive ozone profiling during western US field campaigns. The stratospheric contribution estimated by CMAQ appears to be much lower than the estimates from these prior studies:

A.O. Langford, R.J. Alvarez II, J. Brioude, R. Fine, M. Gustin, J.S. Holloway, M.Y. Lin, R.D. Marchbanks, R.B. Pierce, S.P. Sandberg, C.J. Senff, A.M. Weickmann, E.J. Williams, *Entrainment of stratospheric air and Asian pollution by the convective boundary layer in the Southwestern U.S.*, J. Geophys. Res., 122 (2), doi:10.1002/2016JD025987, 2017.

Lin M., A. M. Fiore, O. R. Cooper, L. W. Horowitz, A. O. Langford, Hiram Levy II, B. J. Johnson, V. Naik, S. J. Oltmans, C. Senff (2012): Springtime high surface ozone events over the western United States: Quantifying the role of stratospheric intrusions, *Journal of Geophysical Research*, 117, D00V22, doi:10.1029/2012JD018151

Langford, A.O., C.J. Senff, R.J. Alvarez II, J. Brioude, O.R. Cooper, J.S. Holloway, M.Y. Lin, R.D. Marchbanks, R.B. Pierce, S.P. Sandberg, A.M. Weickmann , E.J. Williams (2015): An overview of the 2013 Las Vegas Ozone Study (LVOS): Impact of stratospheric intrusions and long-range transport on surface air quality. Atmos. Environ, doi:10.1016/j.atmosenv.2014.08.040

Langford, A. O., Senff, C. J., Alvarez II, R. J., Aikin, K. C., Baidar, S., Bonin, T. A., Brewer, W. A., Brioude, J., Brown, S. S., Burley, J. D., Caputi, D. J., Conley, S. A., Cullis, P. D., Decker, Z. C. J., Evan, S., Kirgis, G., Lin, M., Pagowski, M., Peischl, J., Petropavlovskikh, I., Pierce, R. B., Ryerson, T. B., Sandberg, S. P., Sterling, C. W., Weickmann, A. W., and Zhang, L.: *The Fires, Asian, and Stratospheric Transport-Las Vegas Ozone Study (FAST-LVOS)*, Atmos. Chem. Phys., https://doi.org/10.5194/acp-2021-690, 2022.

Below is additional discussion of the stratospheric  $O_3$  contribution in the EQUATES CMAQ simulations that has been added in Section 3.3. In these additions, the comparisons to previous work are mostly for modeling studies that have reported seasonal mean stratospheric contributions rather than work from field intensives that tend to focus on specific stratospheric intrusion events that are not directly comparable to the seasonal estimates.

"In the 12 km EQUATES simulations, the STRAT  $O_3$  tracer averages 14 ppb in the western US during spring, with a maximum spring average across all western US grid cells of 17 ppb. Using the bias correction approach developed here, we find that the spring average STRAT O3 in the western US is biased low by 3.5 ppb, resulting in an adjusted (i.e., bias corrected) estimate of western US spring average STRAT O3 of 17 ppb. Consistent with the low bias in stratospheric O3 suggested here, other CTMs have estimated higher stratospheric O3 contributions compared to those simulated here with CMAQ. The spring average of stratospheric O3 contributions estimated with the AM3 model has been estimated at 20-25 ppb (Lin et al., 2012a; Langford et al., 2015; Lin et al., 2015). The AM3 estimates of stratospheric O3 have sometimes been estimated to be biased high (Lin et al., 2012a) and have also been shown to lead to overestimated springtime O3 concentrations when used as boundary conditions for regional-scale CMAQ simulations (Hogrefe et al., 2018) but at other times have been estimated to be relatively unbiased based on evaluation against observations from intensive field studies (Langford et al., 2015). The stratospheric O3 contribution simulated by AM3 has been previously found to be higher than that of the GEOS-Chem global model (Fiore et al., 2014). Using GEOS-Chem, Zhang et al. (2014) found the spring mean stratospheric O3 influence in the Intermountain West to range from 8-10 ppb as estimated using the standard GEOS-Chem definition of stratospheric  $O_3$  as described in Zhang et al. (2011) and, alternatively, found a spring mean of 12-18 ppb using a definition of stratospheric O3 adopted from Lin et al. (2012a) (the same method used for the AM3 estimates reported here). Itahashi et al. (2020) previously found that the stratospheric O3 representation in CMAQ was biased low in the free troposphere and suggested that improvements were needed to the CMAQ representation of stratosphere to troposphere transport. Our bias adjusted estimate of western US spring mean stratospheric O3 (17 ppb) falls in between the estimates from the default GEOS-Chem representation (8-10 ppb) and from AM3 (20-25 ppb). As these are seasonal averages, the values are more representative of the continual entrainment of stratospheric air into the troposphere rather than episodic deep stratospheric intrusion events."

**Other comments:**

Lines 29-30: "USB O3 ... is a larger portion of total observed O3 as anthropogenic precursor emissions decline". This statement needs a few references, such as Lin et al. (2017):

Lin, M., W. Horowitz, R. Payton, A.M. Fiore, G. Tonnesen (2017). US surface ozone trends and extremes from 1980 to 2014: Quantifying the roles of rising Asian emissions, domestic controls, wildfires, and climate. Atmos. Chem. Phys., doi:10.5194/acp-17-2943-2017

We have added citations for this sentence, including the one suggested by the reviewer.

Lines 38-48: Need references. Could also discuss the difficulty to separate the anthropogenic and natural driver of wildfire impacts on ozone air quality, as ozone production is enhanced due to mixing of wildfire VOC emissions with urban NOx?

Several references have been added throughout this paragraph. The point about wildfire impacts that the reviewer raises is also relevant to this passage, so the following has been added to this section:

"Wildfires are treated as USB  $O_3$  sources, but the impacts of wildfires on  $O_3$  can be affected by US anthropogenic emissions when VOCs from fires are transported over NOx-rich urban areas, leading to enhanced  $O_3$  production (Jaffe et al., 2013; Langford et al., 2023; Rickly et al., 2023)."

Table 1: The referee agrees with Referee #1 that the authors should list more detailed information regarding model version, simulations types, horizontal and vertical resolution, US anthropogenic emissions, international emissions, fire emissions (including temporal frequency and injection height), and other natural emissions.

Many of these details are available from Tables S4 and S5. Information on the vertical layer structure and more details on the modeling domains have now been added to Table S5. We have also added additional details given below in Section 2.1 of the manuscript. (Note this response is identical to the response to reviewer #1 comment marked "2. Methods")

**Vertical structure:**

[revised manuscript text omitted]

For all of the figures, please indicate in the figure caption whether MDA8 O3 or 24-h O3 is shown. There are some discussions of the metric in the first paragraph of Section 2.1. But it is much easier for readers if you label them as "MDA8 O3" directly in the figure captions.

Captions have been updated to indicate MDA8 O3.

Figures 2 to 5: Results in the maps look pretty similar in their current form. Please use a different colorbar so that the spatial distribution of different model configurations can be better illustrated!

Main text and SI figures have been updated to use different color scales.

---

## Author Response (AR2)

We would like to thank the reviewers for their comments. Below are responses to the individual comments. Responses are in blue, and revised text in the manuscript is in red.

Per the editor, one reviewer has raised the following concern regarding competing interests:

I believe that funding from Phillips 66 - an oil company - for TNS and AGR is a competing interest (conflict of interest) despite the authors not stating so. Phillips 66 is absolutely not a neutral funder, they are - via the American Petroleum Institute - actively lobbying against lowering ozone standards, so I don't see how this is not a major conflict of interest.

The authors disagree with the reviewer that the funding from the Phillips 66 Company constitutes a conflict of interest on the part of some of the authors. We have updated the funding acknowledgements to the following (new text in red):

"TNS and AGR received funding from the Phillips 66 Company. AGR also received funding from NASA HAQAST. Funding organizations have not dictated the topic or content of this work nor have they had any editorial role. The views expressed in this paper are those of the authors and do not necessarily represent the view or policies of the U.S. Environmental Protection Agency, the Phillips 66 Company, or NASA."

**Reviewer 3:**

I was not a reviewer for the original draft, and thus will only make overarching comments.

I found the acronyms of the model runs to be very confusing and not at all intuitive. For that reason, this manuscript was very hard to follow. I highly recommend the authors spell the words out rather than use acronyms. NAT=Natural, LINTL=Long-range international CANMEX=Canada-Mexico USB =U.S. Background, USA=U.S. Anthropogenic (which is different than USA the country, so confusing!) Please spell out these acronyms.

We have updated the acronyms as listed below and no longer use the acronyms in the text. There are many resulting changes in tables, figures, captions, and the text which are not detailed here.

USB becomes US Background

USA becomes US Anthropogenic

INTL becomes International

LINTL becomes Long-range international

CANMEX becomes Canada+Mexico

STRAT becomes Stratospheric

USB_NOSTRAT becomes non-Stratospheric US Background

There is a mismatch between the annual/seasonal averages of MDA8 and how the MDA8 is regulated which is the 4th highest concentration (essentially 99th percentile of annual concentrations). This is not a

relevant comment in all aspects of the manuscript, however, this is particularly relevant for Tables 2 & 3 and Figures 2-5: the U.S. anthropogenic component, ~10 ppb, which makes sense for an annual average, but not relevant for a 99th percentile day. If I had reviewed the original manuscript, I would've had you plot the 99th percentile of Figure 2 and insert/discuss between Figures 2 & 3. That is still my recommendation, although I realize I did not review this in Round 1, so incorporating this comment may take a lot of effort. I will ultimately leave that decision up to the Editor. At the absolute minimum it would be important to acknowledge several times throughout the manuscript that the annual average is not relevant to the current NAAQS, especially near the text referring to Tables 2 & 3, Figures 2-5 (more than a brief mention in Line 764 Track Changes version at the end of the manuscript).

We have added two new figures similar to what is suggested by the reviewer. The new figures show the fourth highest simulated MDA8 $O_3$ in the base case along with the contributions from each of the $O_3$ components on the same day for the PA (Figure 4) and EQUATES (Figure 7) simulations. Additional discussion of these results has also been added to the manuscript.

The biases in the Spring and Summer (Lines 814-816 Tracked Changes version) seem like they warrant further investigation, especially since O3 is regulated based on the 4th highest value which typically occurs in May-Sept. I think Lines 817-832 suggests stratospheric influence is underestimated and missing particulate nitrate photolysis, but I think there may be other reasons. There is no mention of O3 produced from lightning and soil in the Discussion section, and I'm sure both have large uncertainties. Also no mention of O3 deposition uncertainties, or ozone production efficiency uncertainties. All of these uncertainties need to be discussed.

We have added the following to the discussion to mention lightning and soil NOx (new text in red):

"The biases associated with long-range international may be misattributed due to the difficulty of the regression model formulation to isolate stratospheric influences from other natural sources such as lightning and soil $NO_x$, wildfires, and biogenic VOC emissions, all of which have a high degree of uncertainty."

We have also added the following to the discussion about uncertainties in processes other than emissions that affect ozone (new text in red):

"Additional future work could take a process-oriented approach rather than the source-oriented approach described here. A process-oriented approach would focus on how different physical and chemical processes (deposition, transport, photochemical activity, etc.) relate to biases in $O_3$ simulations. The role of uncertainties in $O_3$ deposition and in $O_3$ production efficiency across various chemical regimes could be examined in a more process-focused analysis."

See also the response to the other reviewer's comment #3 where additional discussion of the role of natural emissions, including lightning and soil NOx, has been added.

**Reviewer 4:**

This study highlights biases as seen in US background ozone and its contribution to total ozone as predicted by two sets of CMAQ simulations, using a model-measurement fusion model utilizing a multivariate least square regression approach. The authors highlight that such a data fusion approach leverages the strengths of different data sources to reduce modeled US background (USB) ozone (O3) biases. The reduction in model USB O3 biases is critical, given the NAAQS for ozone will become more stringent, increasing the importance of background O3 (and its sources) to total O3. The manuscript is quite comprehensive and acceptable for publication, given the following main conceptual comments are addressed:

Comments:
1) Abstract + Lines 61-63: "We extend the bias correction method to estimate biases in separate components of USB O3. Separating the USB O3 components provides new insights into the inferred CTM error in USB O3 that was not possible when USB O3 was treated as a lumped quantity". Comments from the erstwhile reviewer(s) to make the abstract more quantitative have been addressed to some extent. However, the findings (or its summary) about 'source-specific US background O3 biases' (or quantitatively how much of them are corrected by this work?) are lacking in the abstract, given the authors propose this manuscript as an update to Skipper et al., 2021 in that specific direction only. Please consider adding 1-2 sentences on the main findings on source-specific USB O3 biases upfront in the abstract. Currently, the variation in USB O3 bias seasonally and with model resolution is quantitatively mentioned in the abstract. Summarize source-specific bias findings as discussed in the conclusions, in the Abstract as well.

As stated in the abstract:

"Correlation among different US background $O_3$ components can increase the uncertainty in the estimation of the source-specific adjustment factors."

And as stated in Section 4:

"The seasonality of inferred long-range international bias highlights a key uncertainty in correlative bias attribution. The biases associated with long-range international may be misattributed due to the difficulty of the regression model formulation to isolate stratospheric influences from other natural sources such as lightning and soil $NO_x$, wildfires, and biogenic VOC emissions, all of which have a high degree of uncertainty. Stratospheric $O_3$ is expected to have similar temporal and spatial patterns to long-range international, with contributions being higher in spring and at high elevations. It is suspected that the regression model formulation may be assigning a negative bias in long-range international to make up for missing stratospheric $O_3$ that has a similar pattern to long-range international while at the same time assigning a high bias for natural to reallocate some of stratospheric $O_3$ that is present in natural to long-range international instead."

For these reasons, reporting quantitatively in the abstract the bias attributed by the regression model approach to natural and long-range international contributions to US background $O_3$ would imply a much greater level of certainty in these numbers than is actually the case. We prefer to keep these in the main text where the quantitative results can be placed in the proper context.

2) Lines 143-145: "The EQUATES hemispheric simulations therefore include losses of O3 over seawater that are not present in the PA hemispheric simulations which could affect O3 transported over the Pacific in particular."
Can you elaborate more on which source-specific US background O3 bias (for example, LINTL? Or some other scenario is more relevant to this?) is this effect the most here (describe using Labels in Table S2)?

The labels in Table S2 were requested to be removed from the text by the other reviewer. We have revised the text as follows:

"One potential source of differences is updates to halogen chemistry introduced in CMAQ v5.3 (Sarwar et al., 2019). These updates in the EQUATES simulations enhance halogen-mediated $O_3$ losses, which are strongest over the oceans. These losses are most relevant for $O_3$ contributions (natural and anthropogenic) that are transported long-distances over oceans."

3) Lines 174-185: "For hemispheric-scale simulations, biogenic VOC emissions are from the Model of Emissions of Gases and Aerosols from Nature version 2.1 (MEGAN2.1) (Guenther et al., 2012). The PA simulations additionally replace MEGAN emissions with emissions from the Biogenic Emission Inventory System (BEIS) (Bash et al., 2016) over North America (US EPA, 2019a).. The EQUATES MEGAN emissions are obtained from a compilation by Sindelarova et al. (2014). Soil NOx emissions for the PA hemispheric simulations are also from MEGAN with replacement by BEIS soil NOx over North America. Soil NOx emissions for the hemispheric EQUATES simulations are from a dataset by the Copernicus Atmosphere Monitoring Service (CAMS, 2018) based on methods by Yienger and Levy (1995). Lightning NO emissions for both the PA and EQUATES hemispheric simulations are from monthly climatology obtained from the Global Emissions Initiative (GEIA) and are based on Price et al. (1997). Lightning NOx was not included in the PA continental-scale simulations, while lightning NOx for the EQUATES continental-scale simulations is calculated using an inline module in CMAQ (Kang et al., 2019)."

In the above manuscript text, the authors highlight the differences in emission inputs for the PA versus EQUATES simulations. Can the authors enunciate/comment more on the impact of these differences on the model ozone biases, as per the scope of the manuscript, say: 1) with (Continental-scale EQUATES) and without Lightning NOx (Continental-scale PA), 2) Different biogenic VOC and NOx (Soil) emissions used in different simulations. For instance, impact of difference in Soil NOx emission inputs on USB O3 bias might be inferred from time series analysis in growing season (say, JJA) in agricultural regions such as, Central Valley and US Midwest.

The impact (or possible impact) of these differences (biogenic emission inputs being different between simulations) needs to be linked more clearly to the findings in the results and/or Conclusions. Also, a small typo here: North America (US EPA, 2019a). (please review and correct of any other such typos throughout the manuscript)

We have added the following to discuss the potential role of the different biogenic VOC, soil NOx, and lightning NOx emission inputs:

"The annual average of simulated US background $O_3$ for the hemispheric-scale (108 km resolution) and continental-scale (12 km resolution) modeling was slightly higher for the EQUATES simulations (32-33 ppb) than for the PA simulations (30-31 ppb). The differences are not explainable by the updated chemical mechanism used in EQUATES because the most relevant updates (halogen-mediated $O_3$ loss) tend to reduce $O_3$ at the northern mid-latitudes (Sarwar et al. 2019; Appel et al., 2021). The difference is

also not likely due to anthropogenic emissions outside of the US, which are similar between the two sets of simulations. So, the higher US background $O_3$ in EQUATES likely relates to differences in the natural emissions. The EQUATES simulations used MEGAN for biogenic emissions throughout the entire Northern Hemisphere while the PA simulations used BEIS for biogenic emissions in North America and MEGAN elsewhere. The two hemispheric model configurations also used different sources for soil $NO_x$ emissions (see Section 2.1) which could contribute to differences in US background $O_3$. Lightning $NO_x$ emissions were the same in EQUATES and PA hemispheric-scale simulations, but the continental-scale PA simulations did not include lightning in the continental domain. Given that US background $O_3$ in both the EQUATES and PA 12 km continental-scale simulations are 1 ppb lower than their northern hemispheric counterparts, the differences in US background $O_3$ in the continental-scale simulations is more likely driven by the large-scale background inherited through the lateral boundary conditions than from differences in lightning $NO_x$ configurations."

---

## Author Response (AR3)

Supplemental Information for

**Source specific bias correction of US background ozone modeled in CMAQ**

T. Nash Skipper[1,*], Christian Hogrefe[2], Barron H. Henderson[2], Rohit Mathur[2], Kristen M. Foley[2], Armistead G. Russell[1]

[1]School of Civil & Environmental Engineering, Georgia Institute of Technology, Atlanta, GA 30332, USA
[2]U.S. Environmental Protection Agency, Research Triangle Park, NC, 27709, USA
[*]Now at U.S. Environmental Protection Agency, Research Triangle Park, NC, 27709, USA

*Correspondence to*: Armistead G. Russell (ar70@gatech.edu)

**Description of CMAQ simulations and O$_3$ components**

**Table S1. Simulation names and descriptions for hemispheric-scale and regional-scale simulations. Table adapted from 2020 O$_3$ Policy Assessment Table 2-1 (US EPA, 2020). Table S1 is reproduced from Table 1 in the main text to aid in interpreting Tables S2 and S3.**

| Simulation | Description |
|---|---|
| BASE | All emission sectors are included. |
| ZUSA | All US anthropogenic emissions are removed including prescribed fires. [a] |
| ZROW | All anthropogenic emissions outside the US are removed including prescribed fires where possible (ROW = rest of world). [b] |
| ZCANMEX | All anthropogenic emissions from Canada and Mexico are removed including prescribed fires where possible. [b] |
| ZANTH | All anthropogenic emissions globally are removed including prescribed fires. [b] |
| STRAT | Tracer species for O$_3$ injected into the upper troposphere/lower stratosphere based on CMAQ potential vorticity parameterization for stratospheric O$_3$. [c] |

[a] Emissions estimated to be associated with intentionally set fires ("prescribed fires") are grouped with anthropogenic fires.

[b] Only for PA simulations

[c] Only for EQUATES simulations.

**Table S2. Expressions used to calculate contributions from specific sources for Policy Assessment simulations described in Table S1. Table adapted from 2020 O$_3$ Policy Assessment Table 2-2 (US EPA, 2020).**

| Label | Name | Description | Expression |
|---|---|---|---|
| BASE | Base | Total Concentration | BASE |
| USB | US Background | US Background | ZUSA |
| USA | US Anthropogenic | US Anthropogenic | BASE – ZUSA |
| INTL | International | Rest of the World Contribution | BASE – ZROW |
| CANMEX | Canada+Mexico | Canada & Mexico Contribution | BASE – ZCANMEX |
| LINTL | Long-range international | Contribution from countries other than the US, Canada, and Mexico | INTL – CANMEX |
| NAT | Natural | Natural Contribution | ZANTH |
| RES-ANTH | Residual anthropogenic | Anthropogenic contribution that is not attributed directly to either the US or International due to non-linear chemistry | BASE – ZANTH – INTL – USA = BASE – ZANTH – (BASE – ZROW) – (BASE – ZUSA) = ZROW + ZUSA – BASE – ZANTH |

**Table S3. Expressions used to calculate contributions from specific sources for EQUATES simulations described in Table S1.**

| Label | Name | Description | Expression |
|---|---|---|---|
| BASE | Total | Total Concentration | BASE |
| USB | US Background | US Background | ZUSA |
| STRAT | Stratospheric | Stratospheric O$_3$ estimate from potential vorticity tracer species | STRAT |
| USB_NOSTRAT | non-Stratospheric US Background | Estimate of USB O$_3$ from sources other than stratospheric O$_3$ | ZUSA – STRAT |

**Table S4. Summary of emissions used for CTM simulations.**

| | PA continental | PA H-CMAQ | EQUATES continental | EQUATES H-CMAQ |
|---|---|---|---|---|
| US anthropogenic | 2016 emissions modeling platform (2016fe) (US EPA, 2019a) | 2016fe | Foley et al. (2023) | Foley et al. (2023) |
| non-US (except Canada and Mexico) | from lateral boundary conditions | EDGAR-HTAP[a] projected to 2014 (US EPA, 2019b) China: Tsinghua University (Zhao et al., 2018) | from lateral boundary conditions | EDGAR-HTAP projected to 2014 China: Tsinghua University |
| Canada and Mexico | 2016fe | 2016fe | Canada: Air Pollutant Emission Inventory by Environment and Climate Change Canada Mexico: Inventory from Mexico's Secretariat of Environment and Natural Resources (SEMARNAT) | Canada: Air Pollutant Emission Inventory by Environment and Climate Change Canada Mexico: Inventory from Mexico's Secretariat of Environment and Natural Resources (SEMARNAT) |
| Lightning | None | GEIA (Price et al., 1997)[b] | CMAQ inline module (Kang et al., 2019) | GEIA |
| Biogenics | Biogenic Emission Inventory System (BEIS) | Model of Emissions of Gases and Aerosols from Nature (MEGAN), except BEIS over North America | BEIS | Hourly CAMS biogenic VOCs v2.2 data (Sindelarova et al., 2014); extension of MEGAN2.1 |
| Soil NOx | BEIS | MEGAN, except BEIS over North America | BEIS | Hourly CAMS soil NO v2.1 data |
| Wildfires | 2016fe | FINNv1.5 (Wiedinmyer et al., 2011), except 2016fe over North America | SMARTFIRE2 + Bluesky | FINNv1.5; SMARTFIRE2 + Bluesky within North America |
| Methane | set to constant value in CMAQ (1850 ppb) | set to constant value in CMAQ (1850 ppb) | set to constant value in CMAQ (1850 ppb) | set to constant value in CMAQ (1850 ppb) |
| Stratospheric $O_3$ | from LBCs, otherwise none | potential vorticity parameterization in CMAQ (Xing et al., 2016; Mathur et al., 2017) | from LBCs, otherwise none | potential vorticity parameterization in CMAQ |

[a] https://edgar.jrc.ec.europa.eu/dataset_htap_v2
[b] https://igacproject.org/activities/GEIA

**Table S5. Summary of model configurations for CTM simulations.**

| | PA continental | PA H-CMAQ | EQUATES continental | EQUATES H-CMAQ |
|---|---|---|---|---|
| CMAQ model version | 5.2.1 | 5.2.1 | 5.3.2 | 5.3.2 |
| Chemical mechanism | cb6r3_ae6nvPOA_aq | cb6r3_ae6_aq | cb6r3_ae7_aq | cb6r3m_ae7_kmtbr |
| Lateral boundary conditions | nested from H-CMAQ to 36 km CMAQ to 12 km CMAQ | clean conditions at equator | Nested from H-CMAQ | clean conditions at equator |
| Meteorology model version | WRF v3.8 | WRF v3.8 | WRF v4.1.1 | WRF v4.1.1 |
| Vertical layers | 35 vertical layers from surface to 50 hPa; surface layer height of approximately 20 m | 44 vertical layers from surface to 50 hPa; surface layer height of approximately 20 m | 35 vertical layers from surface to 50 hPa; surface layer height of approximately 20 m | 44 vertical layers from surface to 50 hPa; surface layer height of approximately 20 m |
| Modeling domains | 396×246 grid cells 12 km domain (12US2); 172×148 grid cells 36 km domain (36US3) | 187×187 grid cells 108 km domain (108NHEMI) | 459×299 12 km domain (12US1) | 187×187 grid cells 108 km domain (108NHEMI) |

**Regression modeling supplemental information**

The regression variables are normalized to zero mean and unit standard deviation. The means and standard deviations for the 2016, 2017, and combined 2016-2017 observations are provided below.

**Table S6. Regression variable means and standard deviations.**

| variable | mean | | | standard deviation | | |
|---|---|---|---|---|---|---|
| | **2016** | **2017** | **2016-2017** | **2016** | **2017** | **2016-2017** |
| lon (°) | -95.4 | -95.0 | -95.2 | 16.0 | 15.7 | 15.8 |
| lat (°) | 37.5 | 37.7 | 37.6 | 4.80 | 4.73 | 4.76 |
| z (m) | 401 | 402 | 402 | 566 | 571 | 569 |
| sin(d) | -0.017 | 0.016 | 0.000 | 0.718 | 0.725 | 0.722 |
| cos(d) | -0.142 | -0.128 | -0.135 | 0.681 | 0.676 | 0.679 |

In the cross-validation summary tables, spatial and temporal withholding refers to randomly assigning 10% of data to the test set, spatial withholding refers to assigning data from 10% of randomly chosen observation sites to the test set, and temporal withholding refers to assigning data from 10% of randomly chosen days of the year to the test set. $O_3$ split refers to the $O_3$ components included in each regression model. The base $O_3$ simulation performance is also provided for comparison to the results of the regression models.

**Table S7. Summary of linear regression model cross-validation root mean square error (RMSE). The performance for the BASE $O_3$ simulations prior to applying the bias adjustment is also provided for comparison. MDA8 $O_3$ components use the acronyms defined in Tables S2 and S3.**

| modelling case | O₃ split | Base Simulation RMSE (ppb) | training RMSE spatial and temporal withholding (ppb) | test RMSE spatial and temporal withholding (ppb) | training RMSE spatial withholding (ppb) | test RMSE spatial withholding (ppb) | training RMSE temporal withholding (ppb) | test RMSE temporal withholding (ppb) |
|---|---|---|---|---|---|---|---|---|
| EQUATES 12 km | USA + USB | 8.09 | 7.25 | 7.25 | 7.25 | 7.22 | 7.25 | 7.28 |
| | USA + USB_NOSTRAT + STRAT | | 7.12 | 7.13 | 7.12 | 7.14 | 7.11 | 7.2 |
| EQUATES 108 km | USA + USB | 9.29 | 8.33 | 8.34 | 8.33 | 8.40 | 8.35 | 8.24 |
| PA 12 km | USA + USB | 8.18 | 7.04 | 7.10 | 7.07 | 6.79 | 7.04 | 7.04 |
| | USA + NAT + INTL | | 7.14 | 7.18 | 7.17 | 6.86 | 7.14 | 7.17 |
| | USA + NAT + LINTL + CANMEX | | 7.09 | 7.13 | 7.12 | 6.82 | 7.09 | 7.09 |
| PA 36 km | USA + USB | 10.04 | 7.96 | 7.97 | 8.01 | 7.47 | 7.97 | 7.89 |
| | USA + NAT + INTL | | 7.98 | 7.98 | 8.02 | 7.55 | 7.98 | 7.93 |
| | USA + NAT + LINTL + CANMEX | | 7.89 | 7.89 | 7.93 | 7.52 | 7.9 | 7.87 |
| PA 108 km | USA + USB | 12.05 | 8.67 | 8.69 | 8.71 | 8.33 | 8.68 | 8.63 |
| | USA + NAT + INTL | | 8.65 | 8.69 | 8.68 | 8.45 | 8.66 | 8.64 |
| | USA + NAT + LINTL + CANMEX | | 8.52 | 8.56 | 8.54 | 8.42 | 8.54 | 8.47 |
| Average | n/a | 9.53 | 7.80 | 7.83 | 7.83 | 7.58 | 7.81 | 7.79 |

**Table S8. Summary of linear regression model cross-validation mean biases (MB). The performance for the base O$_3$ simulations prior to applying the bias adjustment is also provided for comparison. MDA8 O$_3$ components use the acronyms defined in Tables S2 and S3.**

| modelling case | O$_3$ split | Base Simulation MB (ppb) | training MB random split (ppb) | test MB random split (ppb) | training MB site split (ppb) | test MB site split (ppb) | training MB time split (ppb) | test MB time split (ppb) |
|---|---|---|---|---|---|---|---|---|
| EQUATES 12 km | USA + USB | -1.83 | -0.08 | -0.07 | -0.07 | -0.4 | -0.08 | 0.4 |
| | USA + USB_NOSTRAT + STRAT | | -0.12 | -0.12 | -0.11 | -0.12 | -0.12 | 0.38 |
| EQUATES 108 km | USA + USB | 0.66 | -0.1 | -0.07 | -0.1 | -0.28 | -0.1 | 0.31 |
| PA 12 km | USA + USB | 0.49 | -0.09 | -0.1 | -0.09 | -0.55 | -0.09 | 0.54 |
| | USA + NAT + INTL | | -0.16 | -0.15 | -0.16 | -0.62 | -0.16 | 0.47 |
| | USA + NAT + LINTL + CANMEX | | -0.15 | -0.14 | -0.15 | -0.62 | -0.15 | 0.52 |
| PA 36 km | USA + USB | 2.16 | -0.24 | -0.28 | -0.25 | -0.74 | -0.24 | 0.31 |
| | USA + NAT + INTL | | -0.29 | -0.31 | -0.29 | -0.83 | -0.29 | 0.23 |
| | USA + NAT + LINTL + CANMEX | | -0.26 | -0.28 | -0.26 | -0.79 | -0.26 | 0.31 |
| PA 108 km | USA + USB | 4.16 | -0.26 | -0.33 | -0.26 | -0.83 | -0.26 | 0.38 |
| | USA + NAT + INTL | | -0.26 | -0.31 | -0.26 | -0.9 | -0.26 | 0.33 |
| | USA + NAT + LINTL + CANMEX | | -0.23 | -0.28 | -0.22 | -0.86 | -0.23 | 0.39 |
| Average | n/a | 1.13 | -0.19 | -0.20 | -0.19 | -0.63 | -0.19 | 0.38 |

**Table S9. Regression model coefficients and standard errors for USA + USB formulation models. MDA8 $O_3$ components use the acronyms defined in Tables S2 and S3.**

| | EQUATES 12 km | EQUATES 108 km | PA 12 km | PA 36 km | PA 108 km |
|---|---|---|---|---|---|
| $\alpha_{0,USA}$ | $1.093 \pm 0.0021$ | $0.951 \pm 0.0026$ | $0.86 \pm 0.0014$ | $0.762 \pm 0.0016$ | $0.658 \pm 0.0017$ |
| $\alpha_{x,USA}$ | $-0.119 \pm 0.0015$ | $-0.108 \pm 0.0023$ | $-0.054 \pm 0.0011$ | $-0.061 \pm 0.0011$ | $-0.037 \pm 0.0013$ |
| $\alpha_{y,USA}$ | $0.075 \pm 0.0016$ | $0.006 \pm 0.002$ | $-0.006 \pm 0.0011$ | $-0.028 \pm 0.0011$ | $0.005 \pm 0.001$ |
| $A_e$ | $0.01 \pm 0.0023$ | $0.064 \pm 0.0028$ | $0.044 \pm 0.0016$ | $0.078 \pm 0.0016$ | $0.141 \pm 0.002$ |
| $\alpha_{sin,USA}$ | $0.094 \pm 0.0017$ | $0.109 \pm 0.002$ | $0.024 \pm 0.0011$ | $0.018 \pm 0.0011$ | $-0.016 \pm 0.0012$ |
| $\alpha_{cos,USA}$ | $0.085 \pm 0.0018$ | $0.184 \pm 0.0022$ | $0.005 \pm 0.0012$ | $0.043 \pm 0.0013$ | $0.074 \pm 0.0014$ |
| $\alpha_{0,USB}$ | $1.05 \pm 0.0006$ | $1.027 \pm 0.0008$ | $1.053 \pm 0.0007$ | $1.062 \pm 0.0008$ | $1.061 \pm 0.0008$ |
| $\alpha_{x,USB}$ | $-0.02 \pm 0.0006$ | $-0.008 \pm 0.0007$ | $0.008 \pm 0.0006$ | $0.029 \pm 0.0007$ | $0.02 \pm 0.0007$ |
| $\alpha_{y,USB}$ | $-0.016 \pm 0.0005$ | $-0.01 \pm 0.0006$ | $0.022 \pm 0.0006$ | $0.016 \pm 0.0007$ | $0.009 \pm 0.0007$ |
| $\alpha_{z,USB}$ | $0.002 \pm 0.0005$ | $-0.001 \pm 0.0007$ | $0.005 \pm 0.0006$ | $0.004 \pm 0.0006$ | $-0.014 \pm 0.0007$ |
| $\alpha_{sin,USB}$ | $0.044 \pm 0.0006$ | $0.036 \pm 0.0007$ | $0.078 \pm 0.0006$ | $0.078 \pm 0.0007$ | $0.089 \pm 0.0007$ |
| $\alpha_{cos,USB}$ | $0.001 \pm 0.0005$ | $-0.041 \pm 0.0006$ | $0.028 \pm 0.0006$ | $0.001 \pm 0.0006$ | $-0.016 \pm 0.0007$ |

**Table S10. Regression model coefficients and standard errors for USA + NAT + INTL formulation models. MDA8 $O_3$ components use the acronyms defined in Tables S2 and S3.**

| | PA 12 km | PA 36 km | PA 108 km |
|---|---|---|---|
| $\alpha_{0,USA}$ | $0.943 \pm 0.0016$ | $0.835 \pm 0.0018$ | $0.74 \pm 0.002$ |
| $\alpha_{x,USA}$ | $-0.028 \pm 0.0012$ | $-0.031 \pm 0.0013$ | $-0.051 \pm 0.0014$ |
| $\alpha_{y,USA}$ | $0.024 \pm 0.0012$ | $-0.032 \pm 0.0012$ | $0.046 \pm 0.0012$ |
| $\alpha_{z,USA}$ | $0.077 \pm 0.0017$ | $0.134 \pm 0.0018$ | $0.178 \pm 0.0022$ |
| $\alpha_{sin,USA}$ | $0.066 \pm 0.0013$ | $0.066 \pm 0.0013$ | $0.026 \pm 0.0015$ |
| $\alpha_{cos,USA}$ | $-0.014 \pm 0.0014$ | $0.062 \pm 0.0015$ | $0.118 \pm 0.0017$ |
| $\alpha_{0,NAT}$ | $1.065 \pm 0.0022$ | $1.107 \pm 0.0025$ | $1.1 \pm 0.0027$ |
| $\alpha_{x,NAT}$ | $-0.044 \pm 0.0019$ | $-0.012 \pm 0.002$ | $0.051 \pm 0.0021$ |
| $\alpha_{y,NAT}$ | $-0.067 \pm 0.0019$ | $-0.022 \pm 0.002$ | $-0.102 \pm 0.002$ |
| $\alpha_{z,NAT}$ | $-0.041 \pm 0.0019$ | $-0.104 \pm 0.0021$ | $-0.085 \pm 0.0021$ |
| $\alpha_{sin,NAT}$ | $0.009 \pm 0.002$ | $-0.01 \pm 0.0022$ | $0.06 \pm 0.0022$ |
| $\alpha_{cos,NAT}$ | $0.103 \pm 0.0022$ | $-0.016 \pm 0.0026$ | $-0.071 \pm 0.0027$ |
| $\alpha_{0,INTL}$ | $1.332 \pm 0.0051$ | $1.248 \pm 0.0056$ | $1.238 \pm 0.0063$ |
| $\alpha_{x,INTL}$ | $0.15 \pm 0.004$ | $0.123 \pm 0.0041$ | $-0.014 \pm 0.0045$ |
| $\alpha_{y,INTL}$ | $0.197 \pm 0.0038$ | $0.114 \pm 0.0037$ | $0.243 \pm 0.0043$ |
| $\alpha_{z,INTL}$ | $0.09 \pm 0.0042$ | $0.203 \pm 0.0045$ | $0.141 \pm 0.0047$ |
| $\alpha_{sin,INTL}$ | $0.154 \pm 0.0043$ | $0.205 \pm 0.0046$ | $0.069 \pm 0.005$ |
| $\alpha_{cos,INTL}$ | $-0.146 \pm 0.0049$ | $0.005 \pm 0.0055$ | $0.074 \pm 0.0059$ |

**Table S11. Regression model coefficients and standard errors for USA + NAT + LINTL + CANMEX formulation models. MDA8 $O_3$ components use the acronyms defined in Tables S2 and S3.**

| | PA 12 km | PA 36 km | PA 108 km |
|---|---|---|---|
| $\alpha_{0,USA}$ | $0.951 \pm 0.0016$ | $0.859 \pm 0.0018$ | $0.771 \pm 0.002$ |
| $\alpha_{x,USA}$ | $-0.034 \pm 0.0012$ | $-0.046 \pm 0.0013$ | $-0.054 \pm 0.0014$ |
| $\alpha_{y,USA}$ | $0.033 \pm 0.0012$ | $-0.008 \pm 0.0012$ | $0.055 \pm 0.0012$ |
| $\alpha_{z,USA}$ | $0.066 \pm 0.0018$ | $0.12 \pm 0.0018$ | $0.187 \pm 0.0022$ |
| $\alpha_{sin,USA}$ | $0.063 \pm 0.0013$ | $0.062 \pm 0.0013$ | $0.009 \pm 0.0014$ |
| $\alpha_{cos,USA}$ | $-0.004 \pm 0.0014$ | $0.085 \pm 0.0016$ | $0.143 \pm 0.0018$ |
| $\alpha_{0,NAT}$ | $1.037 \pm 0.0023$ | $1.047 \pm 0.0027$ | $1.006 \pm 0.003$ |
| $\alpha_{x,NAT}$ | $-0.043 \pm 0.002$ | $0.014 \pm 0.0021$ | $0.056 \pm 0.0021$ |
| $\alpha_{y,NAT}$ | $-0.073 \pm 0.0019$ | $-0.065 \pm 0.0021$ | $-0.087 \pm 0.002$ |
| $\alpha_{z,NAT}$ | $-0.03 \pm 0.002$ | $-0.082 \pm 0.0022$ | $-0.1 \pm 0.0021$ |
| $\alpha_{sin,NAT}$ | $0.013 \pm 0.002$ | $0.006 \pm 0.0022$ | $0.083 \pm 0.0022$ |
| $\alpha_{cos,NAT}$ | $0.082 \pm 0.0023$ | $-0.056 \pm 0.0027$ | $-0.135 \pm 0.0029$ |
| $\alpha_{0,LINTL}$ | $1.54 \pm 0.0068$ | $1.601 \pm 0.0077$ | $1.822 \pm 0.0085$ |
| $\alpha_{x,LINTL}$ | $0.192 \pm 0.0046$ | $0.121 \pm 0.005$ | $0.095 \pm 0.005$ |
| $\alpha_{y,LINTL}$ | $0.224 \pm 0.0047$ | $0.264 \pm 0.0051$ | $0.151 \pm 0.0052$ |
| $\alpha_{z,LINTL}$ | $0.017 \pm 0.0047$ | $0.104 \pm 0.0053$ | $0.15 \pm 0.0049$ |
| $\alpha_{sin,LINTL}$ | $0.148 \pm 0.0052$ | $0.117 \pm 0.0058$ | $-0.102 \pm 0.0059$ |
| $\alpha_{cos,LINTL}$ | $-0.095 \pm 0.0059$ | $0.063 \pm 0.0066$ | $0.104 \pm 0.0068$ |
| $\alpha_{0,CANMEX}$ | $0.943 \pm 0.0079$ | $0.803 \pm 0.0081$ | $0.667 \pm 0.009$ |
| $\alpha_{x, CANMEX}$ | $0.191 \pm 0.0079$ | $0.135 \pm 0.0068$ | $-0.143 \pm 0.0098$ |
| $\alpha_{y, CANMEX}$ | $0.117 \pm 0.0063$ | $0.004 \pm 0.0052$ | $0.173 \pm 0.0075$ |
| $\alpha_{z, CANMEX}$ | $0.295 \pm 0.0071$ | $0.352 \pm 0.0071$ | $0.248 \pm 0.0085$ |
| $\alpha_{sin, CANMEX}$ | $0.007 \pm 0.0075$ | $0.056 \pm 0.0074$ | $0.021 \pm 0.0082$ |
| $\alpha_{cos, CANMEX}$ | $-0.327 \pm 0.0077$ | $-0.174 \pm 0.008$ | $0.094 \pm 0.0085$ |

**Table S12. Regression model coefficients and standard errors for USA + USB_NOSTRAT + NOSTRAT formulation model. MDA8 $O_3$ components use the acronyms defined in Tables S2 and S3.**

| | EQUATES 12 km |
|---|---|
| $\alpha_{0,USA}$ | $1.088 \pm 0.0015$ |
| $\alpha_{x,USA}$ | $-0.1 \pm 0.0011$ |
| $\alpha_{y,USA}$ | $0.043 \pm 0.0011$ |
| $\alpha_{z,USA}$ | $0.006 \pm 0.0016$ |
| $\alpha_{sin,USA}$ | $0.066 \pm 0.0011$ |
| $\alpha_{cos,USA}$ | $0.062 \pm 0.0013$ |
| $\alpha_{0,USB\_NOSTRAT}$ | $1.058 \pm 0.0017$ |
| $\alpha_{x,USB\_NOSTRAT}$ | $0.097 \pm 0.0012$ |
| $\alpha_{y,USB\_NOSTRAT}$ | $-0.011 \pm 0.001$ |
| $\alpha_{z,USB\_NOSTRAT}$ | $-0.001 \pm 0.0013$ |
| $\alpha_{sin,USB\_NOSTRAT}$ | $0.028 \pm 0.0012$ |
| $\alpha_{cos,USB\_NOSTRAT}$ | $-0.116 \pm 0.0015$ |
| $\alpha_{0,STRAT}$ | $1.038 \pm 0.0022$ |
| $\alpha_{x, STRAT}$ | $-0.167 \pm 0.0015$ |
| $\alpha_{y, STRAT}$ | $-0.035 \pm 0.0013$ |
| $\alpha_{z, STRAT}$ | $0.012 \pm 0.0015$ |
| $\alpha_{sin, STRAT}$ | $0.074 \pm 0.0016$ |
| $\alpha_{cos, STRAT}$ | $0.154 \pm 0.0019$ |

**Simulated O₃ concentrations**

**Table S13. Summary of seasonal average of MDA8 O₃ components for the Policy Assessment set of simulations. Averages are shown for all of the US and separately for the eastern and western US with a longitude of 97 °W serving as the east-west dividing line. The mean across all grid cells within the given area is shown. Numbers in the table are in units of ppb. MDA8 O₃ components use the acronyms defined in Tables S2 and S3.**

| | BASE | USA | NAT | LINTL | CANMEX |
|---|---|---|---|---|---|
| **PA 12 km** | | | | | |
| **DJF** | | | | | |
| all US | 31 | 5 | 17 | 6 | 1 |
| eastern US | 29 | 5 | 15 | 5 | 1 |
| western US | 32 | 4 | 18 | 7 | 1 |
| **MAM** | | | | | |
| all US | 42 | 10 | 21 | 7 | 2 |
| eastern US | 42 | 13 | 19 | 6 | 2 |
| western US | 42 | 7 | 22 | 8 | 2 |
| **JJA** | | | | | |
| all US | 45 | 14 | 22 | 4 | 2 |
| eastern US | 44 | 19 | 18 | 3 | 1 |
| western US | 46 | 10 | 25 | 6 | 2 |
| **SON** | | | | | |
| all US | 40 | 11 | 20 | 5 | 1 |
| eastern US | 40 | 14 | 18 | 4 | 1 |
| western US | 40 | 8 | 22 | 6 | 2 |
| **PA 36 km** | | | | | |
| **DJF** | | | | | |
| all US | 31 | 5 | 16 | 6 | 1 |
| eastern US | 29 | 6 | 15 | 5 | 1 |
| western US | 32 | 4 | 18 | 7 | 1 |
| **MAM** | | | | | |
| all US | 43 | 10 | 21 | 7 | 2 |
| eastern US | 43 | 14 | 19 | 6 | 2 |
| western US | 42 | 8 | 22 | 8 | 2 |
| **JJA** | | | | | |
| all US | 46 | 15 | 22 | 4 | 2 |
| eastern US | 45 | 21 | 18 | 2 | 1 |
| western US | 47 | 11 | 25 | 5 | 3 |
| **SON** | | | | | |
| all US | 41 | 11 | 20 | 5 | 2 |
| eastern US | 41 | 15 | 18 | 4 | 1 |
| western US | 40 | 8 | 22 | 6 | 2 |
| **PA 108 km** | | | | | |
| **DJF** | | | | | |
| all US | 32 | 6 | 17 | 6 | 1 |
| eastern US | 31 | 7 | 15 | 5 | 1 |
| western US | 33 | 5 | 18 | 7 | 1 |
| **MAM** | | | | | |

| | | | | | |
|---|---|---|---|---|---|
| all US | 44 | 11 | 22 | 7 | 2 |
| eastern US | 45 | 15 | 20 | 5 | 2 |
| western US | 43 | 8 | 23 | 8 | 2 |
| **JJA** | | | | | |
| all US | 50 | 16 | 25 | 4 | 2 |
| eastern US | 49 | 22 | 21 | 2 | 1 |
| western US | 50 | 11 | 29 | 5 | 3 |
| **SON** | | | | | |
| all US | 43 | 12 | 21 | 4 | 2 |
| eastern US | 43 | 17 | 20 | 3 | 1 |
| western US | 42 | 8 | 23 | 6 | 2 |

[Figure]

**Figure S1. Seasonal average MDA8 O₃ from Policy Assessment CMAQ simulations. Results are shown for 36 km horizontal resolution for winter (DJF), spring (MAM), summer (JJA), and fall (SON). O₃ concentrations include total (base) O₃ as well as O₃ components from US anthropogenic, natural, long-range international, and Canada+Mexico sources.**

[Figure]

**Figure S2. Seasonal average MDA8 O₃ from Policy Assessment CMAQ simulations. Results are shown for 108 km horizontal resolution for winter (DJF), spring (MAM), summer (JJA), and fall (SON). O₃ concentrations include total (base) O₃ as well as O₃ components from US anthropogenic, natural, long-range international, and Canada+Mexico sources.**

[Figure]

**Figure S3. Average of MDA8 O₃ over the days of the top ten highest base MDA8 O₃ values in the base case from Policy Assessment CMAQ simulations. Results are shown for 12 km (top row), 36 km (middle row), and 108 km (bottom row) horizontal resolutions. O₃ concentrations include total (base) O₃ as well as O₃ components from US anthropogenic, natural, long-range international, and Canada+Mexico sources.**

**Table S14. Summary of seasonal average of MDA8 O₃ components for the EQUATES set of simulations. Averages are shown for all of the US and separately for the eastern and western US with a longitude of 97 °W serving as the east-west dividing line. The mean across all grid cells within the given area is shown. Numbers in the table are in units of ppb. MDA8 O₃ components use the acronyms defined in Tables S2 and S3.**

| | BASE | USA | USB | USB_NOSTRAT | STRAT |
|---|---|---|---|---|---|
| **EQUATES 12 km** | | | | | |
| **DJF** | | | | | |
| all US | 35 | 3 | 32 | 18 | 13 |
| eastern US | 33 | 4 | 30 | 17 | 12 |
| western US | 36 | 3 | 33 | 19 | 14 |
| **MAM** | | | | | |
| all US | 40 | 7 | 33 | 21 | 13 |
| eastern US | 40 | 9 | 31 | 19 | 12 |
| western US | 41 | 5 | 35 | 22 | 14 |
| **JJA** | | | | | |
| all US | 44 | 10 | 33 | 15 | 18 |
| eastern US | 42 | 14 | 28 | 13 | 15 |
| western US | 45 | 7 | 38 | 18 | 20 |
| **SON** | | | | | |
| all US | 37 | 7 | 30 | 16 | 14 |
| eastern US | 37 | 10 | 27 | 14 | 13 |
| western US | 38 | 5 | 33 | 17 | 16 |
| **EQUATES 108 km** | | | | | |
| **DJF** | | | | | |
| all US | 36 | 4 | 32 | --- | --- |
| eastern US | 34 | 4 | 30 | --- | --- |
| western US | 38 | 4 | 34 | --- | --- |
| **MAM** | | | | | |
| all US | 42 | 8 | 34 | --- | --- |
| eastern US | 42 | 10 | 32 | --- | --- |
| western US | 42 | 6 | 36 | --- | --- |
| **JJA** | | | | | |
| all US | 46 | 12 | 35 | --- | --- |
| eastern US | 46 | 16 | 29 | --- | --- |
| western US | 47 | 8 | 39 | --- | --- |
| **SON** | | | | | |
| all US | 39 | 8 | 31 | --- | --- |
| eastern US | 38 | 10 | 28 | --- | --- |
| western US | 39 | 6 | 33 | --- | --- |

[Figure]

**Figure S4. Seasonal average MDA8 O₃ from EQUATES CMAQ simulations. Results are shown for 12 km horizontal resolution for winter (DJF), spring (MAM), summer (JJA), and fall (SON). O₃ concentrations include total (base) O₃ as well as O₃ components from US anthropogenic and US background sources.**

[Figure]

**Figure S5. Seasonal average MDA8 O₃ from EQUATES CMAQ simulations. Results are shown for 108 km horizontal resolution for winter (DJF), spring (MAM), summer (JJA), and fall (SON). O₃ concentrations include total (base) O₃ as well as O₃ components from US anthropogenic and US background sources.**

[Figure]

**Figure S6. Average of MDA8 O₃ over the days of the top ten highest base MDA8 O₃ values from EQUATES CMAQ simulations. Results are shown for 12 km resolution (top and middle rows) and 108 km (bottom row). O₃ concentrations include total (base) O₃ as well as O₃ components from US anthropogenic, non-stratospheric US background, and stratospheric sources for 12 km. For both the 12 km and 108 km simulations, O₃ concentrations of base, US anthropogenic, and total US background are also shown.**

**Seasonal average inferred O₃ model biases**

**Table S15. Summary of annual and seasonal average of MDA8 O₃ component inferred biases for the Policy Assessment set of simulations. Averages are shown for all of the US and separately for the eastern and western US with a longitude of 97 °W serving as the east-west dividing line. The mean across all grid cells within the given area is shown. Numbers in the table are in units of ppb. MDA8 O₃ components use the acronyms defined in Tables S2 and S3.**

| | BASE | USA | NAT | LINTL | CANMEX |
|---|---|---|---|---|---|
| **PA 12 km** | | | | | |
| **annual** | | | | | |
| all US | -0.8 | 0.3 | -0.2 | -3.2 | -0.2 |
| eastern US | 0.6 | 1.3 | -0.4 | -3.0 | 0.0 |
| western US | -1.9 | -0.5 | -0.1 | -3.4 | -0.4 |
| **DJF** | | | | | |
| all US | -2.9 | 0.0 | -2.2 | -3.1 | 0.5 |
| eastern US | -2.3 | 0.4 | -2.2 | -3.2 | 0.6 |
| western US | -3.4 | -0.4 | -2.3 | -3.0 | 0.3 |
| **MAM** | | | | | |
| all US | -4.0 | -0.5 | -0.2 | -5.5 | -0.3 |
| eastern US | -2.6 | 0.2 | -0.3 | -5.2 | -0.1 |
| western US | -5.2 | -1.1 | -0.1 | -5.9 | -0.5 |
| **JJA** | | | | | |
| all US | 1.8 | 0.6 | 2.1 | -2.6 | -1.0 |
| eastern US | 3.8 | 2.1 | 1.7 | -1.9 | -0.6 |
| western US | 0.1 | -0.5 | 2.5 | -3.2 | -1.3 |
| **SON** | | | | | |
| all US | 2.0 | 1.2 | -0.6 | -1.6 | 0.0 |
| eastern US | 3.3 | 2.5 | -0.7 | -1.6 | 0.1 |
| western US | 0.9 | 0.1 | -0.5 | -1.6 | -0.1 |
| **PA 36 km** | | | | | |
| **annual** | | | | | |
| all US | 0.4 | 1.3 | 1.0 | -4.4 | 0.0 |
| eastern US | 2.0 | 3.3 | -1.0 | -3.1 | 0.3 |
| western US | -1.0 | -0.3 | 2.6 | -5.5 | -0.3 |
| **DJF** | | | | | |
| all US | -1.7 | -0.4 | 1.9 | -5.5 | 0.3 |
| eastern US | -1.2 | 0.3 | 0.1 | -4.3 | 0.6 |
| western US | -2.1 | -1.0 | 3.4 | -6.5 | 0.1 |
| **MAM** | | | | | |
| all US | -3.1 | 0.5 | 0.4 | -6.4 | -0.1 |
| eastern US | -1.4 | 2.1 | -1.7 | -4.8 | 0.2 |
| western US | -4.5 | -0.8 | 2.1 | -7.6 | -0.4 |
| **JJA** | | | | | |
| all US | 3.0 | 3.4 | 0.0 | -2.7 | -0.4 |

| | | | | | |
|---|---|---|---|---|---|
| eastern US | 6.0 | 6.8 | -2.1 | -1.4 | 0.0 |
| western US | 0.4 | 0.7 | 1.7 | -3.8 | -0.8 |
| **SON** | | | | | |
| all US | 3.3 | 1.7 | 1.6 | -3.1 | 0.1 |
| eastern US | 4.6 | 3.9 | -0.5 | -2.1 | 0.4 |
| western US | 2.2 | 0.0 | 3.3 | -4.0 | -0.1 |
| **PA 108 km** | | | | | |
| **annual** | | | | | |
| all US | 2.1 | 2.3 | 2.4 | -5.2 | 0.2 |
| eastern US | 4.1 | 5.5 | -0.9 | -3.4 | 0.6 |
| western US | 0.4 | -0.4 | 5.3 | -6.8 | -0.1 |
| **DJF** | | | | | |
| all US | -0.5 | -0.3 | 4.4 | -6.6 | 0.1 |
| eastern US | 0.0 | 1.0 | 1.5 | -4.9 | 0.4 |
| western US | -0.9 | -1.4 | 6.9 | -8.1 | -0.1 |
| **MAM** | | | | | |
| all US | -1.5 | 2.1 | -0.7 | -5.8 | 0.4 |
| eastern US | 0.9 | 5.1 | -3.8 | -3.9 | 0.8 |
| western US | -3.5 | -0.4 | 2.1 | -7.4 | -0.1 |
| **JJA** | | | | | |
| all US | 5.1 | 5.6 | 0.1 | -3.3 | 0.3 |
| eastern US | 8.5 | 10.8 | -3.6 | -1.5 | 0.6 |
| western US | 2.1 | 1.0 | 3.3 | -4.9 | 0.1 |
| **SON** | | | | | |
| all US | 5.4 | 1.9 | 5.8 | -5.3 | 0.1 |
| eastern US | 6.8 | 4.9 | 2.2 | -3.5 | 0.4 |
| western US | 4.1 | -0.7 | 9.0 | -6.9 | -0.2 |

[Figure]

**Figure S7. Seasonal average of inferred MDA8 O₃ model bias from 12 km horizontal resolution Policy Assessment CMAQ simulations. O₃ concentrations include total (base) O₃ as well as O₃ components from US anthropogenic, natural, long-range international, and Canada+Mexico sources.**

[Figure]

**Figure S8. Seasonal average of inferred MDA8 O₃ model bias from 36 km horizontal resolution Policy Assessment CMAQ simulations. O₃ concentrations include total (base) O₃ as well as O₃ components from US anthropogenic, natural, long-range international, and Canada+Mexico sources.**

[Figure]

**Figure S9. Seasonal average of inferred MDA8 O$_3$ model bias from 108 km horizontal resolution Policy Assessment CMAQ simulations. O$_3$ concentrations include total (base) O$_3$ as well as O$_3$ components from US anthropogenic, natural, long-range international, and Canada+Mexico sources.**

**Table S16. Summary of annual and seasonal average of MDA8 O₃ component inferred biases for the EQUATES set of simulations. Averages are shown for all of the US and separately for the eastern and western US with a longitude of 97 °W serving as the east-west dividing line. The mean across all grid cells within the given area is shown. Numbers in the table are in units of ppb. MDA8 O₃ components use the acronyms defined in Table S2.**

[revised manuscript text omitted]